# Toll-like receptor signaling in thymic epithelium controls monocyte-derived dendritic cell recruitment and Treg generation

Matouš Vobořil[1], Tomáš Brabec[1], Jan Dobeš[1], Iva Šplíchalová[1], Jiří Březina[1], Adéla Čepková[1], Martina Dobešová[1], Aigerim Aidarova[1], Jan Kubovčiak[2], Oksana Tsyklauri[3], Ondřej Štěpánek[3], Vladimír Beneš[4], Radislav Sedláček[5], Ludger Klein[6], Michal Kolář[2] & Dominik Filipp[1✉]

The development of thymic regulatory T cells (Treg) is mediated by Aire-regulated self-antigen presentation on medullary thymic epithelial cells (mTECs) and dendritic cells (DCs), but the cooperation between these cells is still poorly understood. Here we show that signaling through Toll-like receptors (TLR) expressed on mTECs regulates the production of specific chemokines and other genes associated with post-Aire mTEC development. Using single-cell RNA-sequencing, we identify a new thymic CD14+Sirpα+ population of monocyte-derived dendritic cells (CD14+moDC) that are enriched in the thymic medulla and effectively acquire mTEC-derived antigens in response to the above chemokines. Consistently, the cellularity of CD14+moDC is diminished in mice with MyD88-deficient TECs, in which the frequency and functionality of thymic CD25+Foxp3+ Tregs are decreased, leading to aggravated mouse experimental colitis. Thus, our findings describe a TLR-dependent function of mTECs for the recruitment of CD14+moDC, the generation of Tregs, and thereby the establishment of central tolerance.

[1] Laboratory of Immunobiology, Institute of Molecular Genetics of the Czech Academy of Sciences, Prague, Czech Republic. [2] Laboratory of Genomics and Bioinformatics, Institute of Molecular Genetics of the Czech Academy of Sciences, Prague, Czech Republic. [3] Laboratory of Adaptive Immunity, Institute of Molecular Genetics of the Czech Academy of Sciences, Prague, Czech Republic. [4] Genomics Core Facility, EMBL, Services & Technology Unit, Heidelberg, Germany. [5] Czech Centre for Phenogenomics & Laboratory of Transgenic Models of Diseases, Institute of Molecular Genetics of the Czech Academy of Sciences, Prague, Czech Republic. [6] Faculty of Medicine, Institute for Immunology, Ludwig-Maximilans-Universitat, Munich, Germany.
✉email: dominik.filipp@img.cas.cz

The establishment of tolerance is a fundamental attribute of a healthy immune system. Since T cell antigen receptors (TCRs) are generated by random somatic recombination, i.e. could be self or nonself-specific, T cells that express a self-reactive TCR must be removed from the conventional T cell repertoire. The critical part of this process occurs in the thymic medulla where the strength of TCR recognition of self-antigens is probed by various types of antigen presenting cells (APCs), mainly dendritic cells (DCs), B-cells, and highly specialized medullary thymic epithelial cells (mTECs)[1]. mTECs mediate the promiscuous expression of thousands of otherwise strict tissue-restricted self-antigens (TRAs), a large number of which are under the control of the transcriptional regulator Aire[2]. The presentation of TRAs by mTECs can result in either the deletion of self-reactive T cells[3] or their conversion into Tregs[4,5].

It has been recently demonstrated that the process of cooperative antigen transfer (CAT) from mTECs to DCs is essential for the establishment of thymic tolerance[6–11]. The complexity of CAT is foremost due to the heterogeneity of DCs in the thymus. These CD11c[+] cells are comprised of two major categories: B220[+] plasmacytoid DCs (pDC) and classical DCs (cDCs), the latter which can be subdivided into Xcr1[+]CD8α[+]Sirpα[−] classical type 1 DCs (cDC1) and Xcr1[−]CD8α[−]Sirpα[+] classical type 2 DCs (cDC2)[12,13]. While cDC1 arise primarily in the thymus, cDC2 and pDCs originate extrathymically and then migrate to the thymic medullary region[14,15]. mTEC-derived antigens are transferred both to thymic resident cDC1[6,10] and cDC2[16,17]. Although it has been shown that the migration of cDC1 and cDC2 to the vicinity of mTECs is affected by a gradient of Xcl1[18] and Ccr2/Ccr7 ligands, respectively[19,20], the potential involvement of other chemokines in the regulation of CAT still awaits resolution.

Toll-like receptors (TLRs) sense various immunologically relevant microbial ligands such as lipoproteins, carbohydrates, and nucleic acids. All TLRs, with the exception of TLR3, signal through the adaptor protein, MyD88, which via the activation of the NF-κB pathway induces the expression of pro-inflammatory cytokines, chemokines, and other inflammation-related molecules[21]. While the exact role of non-canonical NF-κB signaling in the development and function of mTECs has been previously demonstrated[22–24], the impact of TLR signaling via the canonical NF-κB pathway in the physiology of mTECs remains undetermined.

Here, we show that, among TLRs, mTECs abundantly express TLR9, and the stimulation of which leads to the influx of Xcr1[−] Sirpα[+] cDC2 into the thymic medulla. RNA sequencing of stimulated mTECs reveals that the mechanism underpinning this phenomenon is related to the upregulation of a set of chemokines, whose receptors are predominantly expressed by a CD14[+] subset of thymic DCs, which have been identified as monocyte-derived DCs (CD14[+]moDC). Furthermore, mice with MyD88-deficient TECs, which exhibit a deficiency in the recruitment of CD14[+] moDC, also suffer from a decreased thymic Treg output and functionality, which renders the peripheral T cell repertoire prone to colitis induction.

## Results

**mTECs express a set of TLRs and signaling adaptors**. The function of TLR signaling in the physiology of mTECs has not yet been studied in detail[25–27]. We first determined that both mTECs[low] and mTECs[high] subsets (Fig. 1a and Supplementary Fig. 1a) expressed TLR2, 3, 4, and 9 (Fig. 1b). Remarkably, TLR9, which recognizes bacterial, viral or altered DNA[21] and ligands associated with cellular stress[28], is highly expressed by mTECs[high] at levels comparable to thymic cDCs (Fig. 1a, b and Supplementary Fig. 1b). Transcripts of TLR adaptors *MyD88* and *Trif*[21] were also readily detectable (Fig. 1c). Although the levels of TLR4

and TLR9 were higher in mTECs[high], the major producers of Aire, our analysis of Aire[+/+] and Aire[−/−] mice revealed that TLRs are expressed in an Aire-independent manner (Fig. 1d).

To assess the significance of TLR/MyD88 signaling in TECs development, we crossed a thymic epithelial cell-specific Foxn1[Cre] driver[29] with a MyD88[fl/fl] transgenic mice[30] (hereafter called MyD88[ΔTEC]). In comparison to the control, MyD88[ΔTECs] mice showed no significant differences in the frequency of all tested TEC subpopulations (Fig. 1e, f), suggesting that canonical NF-κB signaling through TLRs/MyD88 does not affect mTEC[high] maturation. Similarly, in all mTEC[high] subsets, the expression of CD80, CD86, PD-L1, CD40, and ICOSL on was not altered (Supplementary Fig. 1c).

Together, this data demonstrates that TLRs are broadly expressed by mTECs and MyD88-dependent signaling has no apparent impact on TEC subpopulation frequency.

**MyD88-dependent chemokine expression in mTECs[high]**. Given the high expression of selected TLRs in mTECs[high] cells, we assessed the impact of the absence of TLR signaling in unperturbed conditions. RNA-sequencing of mTECs[high] (sorted as shown in Supplementary Fig. 1a) from wild type (MyD88[fl/fl]) and MyD88[ΔTECs] mice revealed MyD88-dependent transcriptional variance (Fig. 2a) defined by 303 differentially expressed transcripts (Fig. 2b and Supplementary Data 1 and 2). While 206 of these transcripts were induced and 97 repressed by MyD88, they were not enriched for Aire-dependent or Aire-independent TRA genes[31] (Supplementary Fig. 2a, left panel). Consistent with the role of TLR/MyD88 signaling in epithelial cells[21], we found several differentially expressed genes (DEGs) which fell into one of two categories: (i) *Il1f6* and *Csf2* cytokines, (ii) *Ccl25*, *Ccl4*, and *Ccl24* chemokines. These mediators act through receptors that are primarily expressed by myeloid cells and DCs[32]. Specifically, IL36R, the receptor for IL1F6, is expressed by DCs and T cells[33] while Csf2r, the receptor for Csf2, is expressed mostly by monocytes, macrophages, and granulocytes[34]. The Ccr9, the receptor for Ccl25, is expressed by both thymocytes and pDCs driving their migration into the thymus[14,35]. Both Ccr5 (receptor for Ccl4) and Ccr3 (receptor for Ccl24) are expressed predominantly on granulocytes and DCs modulating their migration into inflamed tissues[32,36]. qRT-PCR analysis confirmed MyD88-regulated expression of selected genes in mTECs[high] (Fig. 2c). Since the TLRs were postulated to sense both microbial and endogenous molecules[21], we examined which of them could potentially act as a trigger. The analysis of mRNA expression of MyD88-dependent cytokines and chemokines (Fig. 2b, c) in the mTEC[high] population isolated from either Germ-free (GF) or specific-pathogen-free (SPF) mice was comparable (Supplementary Fig. 2b), indicating that these signals are likely of endogenous origin.

Next, we assessed the response of mTECs to TLR/ MyD88 stimulation. Given the high expression of TLR9 (Fig. 1b), we stimulated mTECs[high] from MyD88-deficient (MyD88[−/−]) and WT (MyD88[+/+]) mice in vitro with CpG oligodeoxynucleotides (CpG ODN) or PBS. RNA-sequencing revealed significant changes in the transcriptional profile only in MyD88[+/+] cells. Notably, 347 DEGs were associated with TLR9 stimulation (Fig. 2d, e and Supplementary Data 3 and 4), and of these, 198 were upregulated while 149 were downregulated. However, the pattern of expression of TRA genes remained largely unchanged after in vitro CpG ODN stimulation (Supplementary Fig. 2a, right panel). Importantly, among the most upregulated DEGs were two sets of chemokines: (i) *Cxcl1*, *2*, *3*, and *5*, which signal via the Cxcr2 receptor, expressed predominantly on neutrophils[37] and (ii) *Ccl3*, *5* and *20* which signal via various chemokine receptors, including Ccr1, 3, 5, 6 which are expressed mostly on myeloid

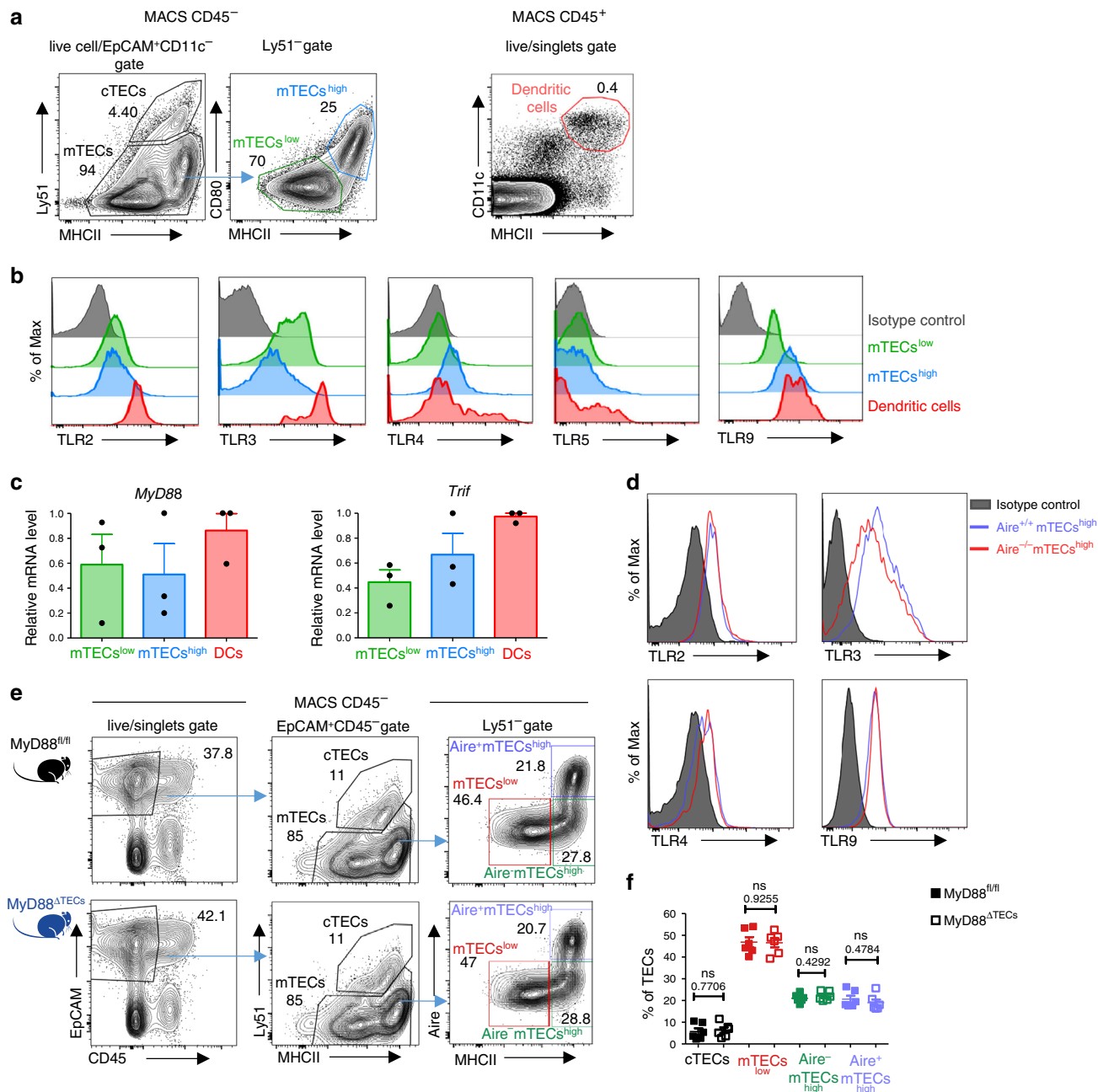

**Fig. 1 mTECs express a set of TLRs and their signaling adaptors independently of Aire. a** Gating strategy used for the analysis of TEC populations and general thymic conventional DCs. MACS enriched CD45⁻ and EpCAM⁺CD11c⁻ pre-gated cells were further divided into cTECs (Ly51⁺), mTECs^low (MHCII^low CD80^low), and mTECs^high (MHCII^high CD80^high). Thymic conventional DCs were gated as CD11c⁺MHCII⁺ from the CD45⁺ fraction. A more detailed gating strategy is found in Supplementary Fig. 1a, b. **b** Representative flow cytometry histograms of TLR expression on mTECs and DCs isolated from the thymus (n = 3 independent experiments). **c** MyD88 and Trif mRNA expression is determined by qRT-PCR from FACS sorted mTECs and DCs. The expression is calculated relative to Casc3 and normalized to the highest value within each experiment=1 (mean ± SEM, n = 3 samples). **d** Representative flow cytometry histograms of TLR expression on mTECs from Aire^+/+ and Aire^−/− mice, (n = 3 independent experiments). **e** Representative comparative flow cytometry plots of different TEC subpopulations in MyD88^fl/fl and MyD88^ΔTECs mice. **f** Quantification of TEC frequencies from plots in e (mean ± SEM, n = 6 mice). Statistical analysis was performed by unpaired, two-tailed Student's t-test, p-values are shown. ns = not significant.

cells[32]. Cytokines (Tnfα, Il-6, Il12a, Il1f6 and Csf2) and other genes (Cd40) were also found to be upregulated (Fig. 2e). The upregulation of Cxcl1 and Ccl5 chemokines after in vitro (Fig. 2f) as well as in vivo intrathymic TLR9 stimulation (Fig. 2g) was confirmed by qRT-PCR analysis. As shown in Supplementary Fig. 2c, repeated intraperitoneal (i.p.) injection of CpG ODN was insufficient for the upregulation of chemokines in mTECs^high. It is of note that in vitro stimulation of TLR4 on mTECs^high by LPS

also resulted in the upregulation of the previously noted chemokines, albeit at a lower level (Supplementary Fig. 2d).

In addition to TLRs, MyD88 also conveys signals generated by IL-1 family cytokines, such as IL-1β, IL-18 or IL-33[38]. Even though the receptors for these cytokines are expressed by mTECs^high (Supplementary Fig. 3a), only in vitro stimulation with IL-1β lead to the upregulation of cytokines and chemokines induced by TLR9 stimulation (Supplementary Fig. 3b).

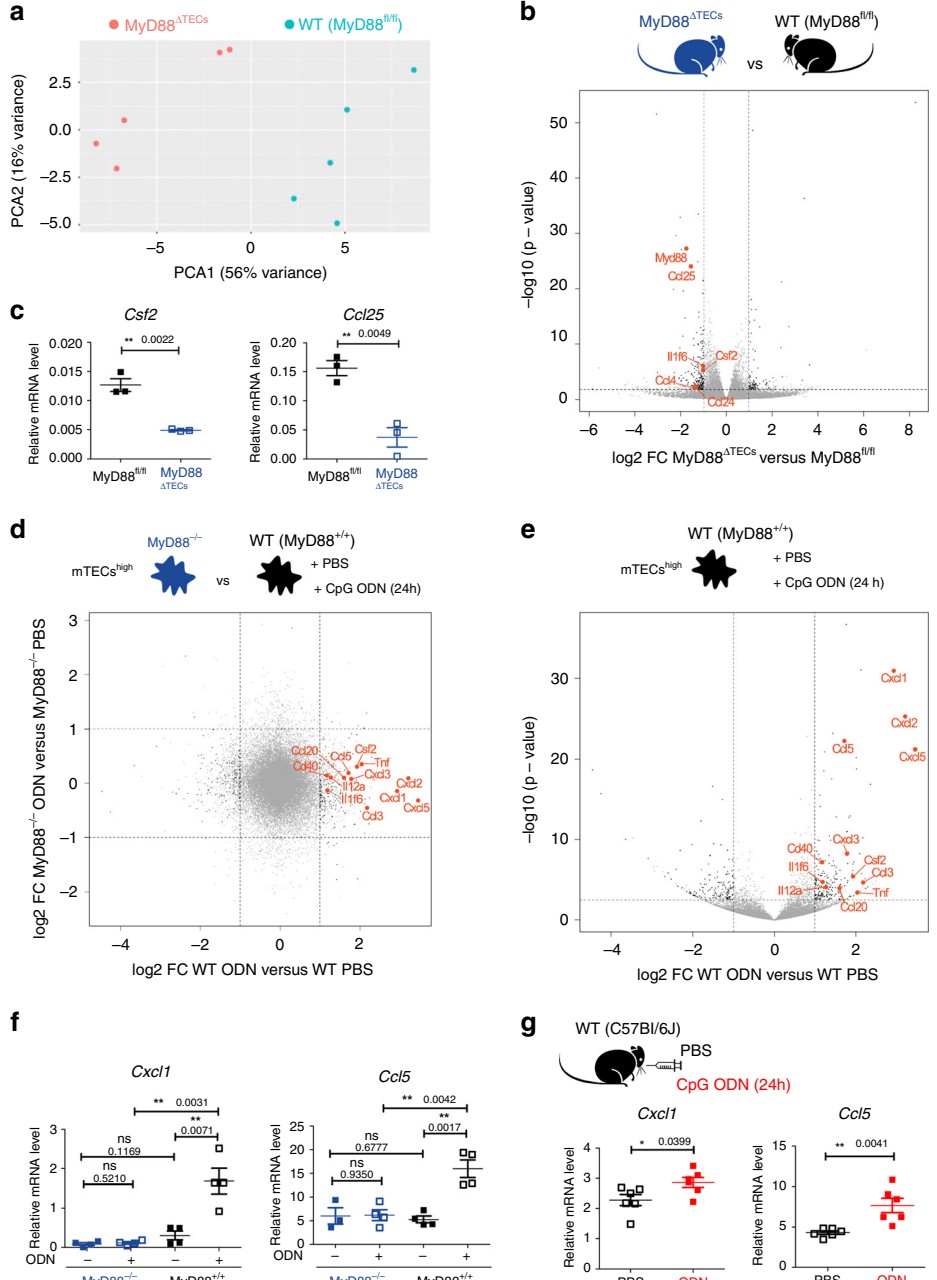

**Fig. 2 TLR/MyD88 signaling in mTECs^high drives the expression of cytokines and chemokines. a** Principal component analysis of bulk RNA-sequencing data from mTECs^high (sorted as in Supplementary Fig. 1a) derived from MyD88^fl/fl and MyD88^ΔTECs mice. Data represents the analysis of $n = 5$ samples for each condition. **b** Volcano plot analysis of RNA-sequencing data described in **a**. Fold-change cutoff of log2 = ±1,0 and $p$-value: 0.05 are marked by dashed lines (also in **d**, **e**). Differentially expressed genes are depicted in black, genes of interest are in red, and other detected genes in grey. **c** qRT-PCR analysis of relative mRNA expression normalized to Casc3 of genes selected from b (mean ± SEM, $n = 3$ samples). **d** Fold-change fold-change plot of RNA-sequencing data from CpG ODN or PBS in vitro stimulated mTECs^high (sorted as in Supplementary Fig. 1a) from MyD88^+/+ and MyD88^−/− mice ($n = 4$ samples for each condition. Color code as in **b**. **e** Volcano plot analysis of RNA-sequencing data from d, comparing CpG ODN versus PBS in vitro stimulated mTECs^high from MyD88^+/+ mice. Statistical analysis for b, d and e was performed by Wald test, $p$-value cutoff: 0.05. **f**, **g** qRT-PCR analysis of Cxcl1 and Ccl5 mRNA expression (normalized to Cacs3) from in vitro (mean ± SEM, $n = 4$ samples) and intrathymically (mean ± SEM, $n = 6$ mice), respectively, CpG ODN or PBS stimulated mTECs^high from indicated animals. Statistical analysis for **c**, **f**, and **g** was performed by unpaired, two-tailed Student's $t$-test, $p \leq 0.05 = *$, $p \leq 0.01 = **$, ns not significant.

Besides chemokines and cytokines, TLR/MyD88 signaling in mTECs^high (Fig. 2b) also regulated the expression of molecules associated with cornified epithelial pathway[39] (Supplementary Data 1–4). This specifically relates to genes that are associated with post-Aire mTECs[40,41], such as *Krt10*, *Krt77* and *Flg2* (Supplementary

Fig. 3c). Moreover, previously published data has shown the enhanced expression of *Il1f6*, *Cxcl3* and *Cxcl5* in post-Aire mTECs[42]. Thus, we enumerated the total numbers of Involucrin^+EpCAM^+ cells in the medullary region of the CpG ODN intrathymically stimulated thymus. We did not observe any changes in the frequency

of general mTECs subsets (Supplementary Fig. 3d) although the total numbers of Involucrin[+] post-Aire mTECs were significantly increased (Supplementary Fig. 3e, f).

Together, these results show that TLR/MyD88 signaling in mTECs under physiological or stimulatory conditions regulates the differentiation of mTEC[high] cells into Involucrin[+] post-Aire stage. This stage is associated with the expression of a set of chemokines that signal via an overlapping set of chemokine receptors that are primarily expressed by DCs[32].

**TLR9/MyD88 signaling in mTECs targets Sirpα[+] cDC2.** Migration of different DC subsets into the thymus is orchestrated by distinct chemokines[14,18,19]. Thus, we next assessed which of these subsets would be the target for TLR9/MyD88-induced chemokines in TECs. We sorted three main subsets of CD11c[+]MHCII[+] thymic DCs: B220[+] pDC, Sirpα[−]Xcr1[+] cDC1, and Sirpα[+]Xcr1[−] cDC2 (Supplementary Fig. 4a), along with Gr-1[+] granulocytes, CD4 and CD8 single positive thymocytes and performed qRT-PCR analysis of the chemokine receptors indicated above. Remarkably, apart from granulocytes, the chemokine receptors $Cxcr2$, $Ccr1$, $3$, $5$, and $6$ were mostly expressed by DCs, specifically by cDC2 and pDC (Fig. 3a). This prompted us to quantify the relative frequencies of all thymic DC subsets in MyD88[ΔTECs] in comparison to WT (MyD88[fl/fl]) mice. In unstimulated conditions, TEC-intrinsic MyD88 signaling did not change the total frequency of CD11c[+]MHCII[+] DCs (Fig. 3b, left plot). However, we observed alterations in the frequencies of DC subsets. While cDC1 were increased, the frequencies of pDC and cDC2 were diminished in the MyD88[ΔTECs] thymus (Fig. 3b). In contrast, FACS analysis of TLR9-stimulated thymi revealed a significant increase in cDC2 accompanied by decreased cDC1 in the thymus of WT (MyD88[fl/fl]) (Fig. 3c and Supplementary Fig. 4b, c) but not MyD88[ΔTECs] animals (Fig. 3c). The frequencies of pDC remained comparable under these two conditions. This demonstrates that the recruitment of cDC2 to the thymus is attributable specifically to TLR9 signaling in TECs (Fig. 3c and Supplementary Fig. 4b). In agreement with medullary localization of cDC2 (Supplementary Fig. 4d), microscopically examined thymi from WT mice stimulated with CpG ODN showed an enrichment of CD11c[+]Sirpα[+] cDC2 exclusively in the keratin14-rich medullary region (Figs. 3d, e).

Together, this data suggests that MyD88-driven chemokines expressed by mTECs[high], target receptors on thymic Sirpα[+] cDC2 and mediate their recruitment to the thymic medulla in steady state and TLR9 stimulatory conditions.

**TLR9/MyD88 signaling in mTECs recruits CD14[+]moDCs.** Chemokine-dependent migration of DCs to the proximity of mTECs, which underpins the mechanisms of CAT[18], has been shown to be essential for the presentation of mTEC-derived antigens by DCs[6,10]. One prediction from the TEC-dependent TLR/MyD88-induced influx of Sirpα[+]cDC2 to the thymic medulla is that the frequency of CAT to this subset would be enhanced.

To verify this prediction, we crossed Foxn1[Cre] mice with ROSA26[TdTOMATO] leading to TEC-specific, cytoplasmic expression of TdTOMATO (TdTOM) protein in the thymus. In agreement with a previous study[9] and as shown in Supplementary Fig. 5a, we found two major populations of TdTOM[+] cells: (i) a TdTOM[high] EpCAM[+] population which was CD45[−] and represented TECs expressing TdTOM endogenously (Supplementary Fig. 5b); and (ii) a CD45[+] TdTOM[+] population comprised of mainly CD11c[+] DCs (Supplementary Fig. 5a) which acquired TdTOM via CAT (Fig. 4a). Interestingly, these DCs were enriched for the EpCAM[+] marker (Fig. 4b) which was likely co-transferred with TdTOM[9]. Bone marrow (BM) chimeras

of lethally irradiated Foxn1[Cre]ROSA26[TdTOMATO] mice reconstituted with WT BM cells showed that around 6% of donor-derived DCs acquired TdTOM (Supplementary Fig. 5c–e). This formally demonstrates that TdTOM is transferred from TECs to DCs.

It has been previously documented that distinct subtypes of thymic DCs vary in their capacity to acquire antigens from TECs[6,10,11,16]. Whereas CAT of TdTOM from TECs to cDC1 and cDC2 is very potent in the Foxn1[Cre]ROSA26[TdTOMATO] system, it is limited in the case of pDC (Fig. 4e, f). This result was also corroborated with the use of BM chimeras which were described above (Supplementary Fig. 5f). Flow cytometry imaging showed that transferred TdTOM in MHCII[+]CD11c[+] DCs is localized intracellularly (Fig. 4g).

To determine the heterogeneity of all thymic DC subsets that participate in CAT, we performed single-cell RNA-sequencing (ddSEQ)[43] of Gr-1[−]CD11c[+]TdTOM[+] cells isolated from thymi of Foxn1[Cre]ROSA26[TdTOMATO] mice. Two-dimensional tSNE projection clustering analysis revealed five different clusters of TdTOM[+] DCs (Fig. 5a). Based on their expression profiles and previously described signature genes of cells from mononuclear phagocyte system (MPS)[44], we designated the clusters in accordance with MPS nomenclature[13]: two cDC1 clusters (Batf3): a cDC1a (Ccl5 and Ccr7) and cDC1b (Cd8a, Itgae, Xcr1 and Ppt1)[45]; cDC2 cluster (Sirpα, Mgl2 and Cd209a)[12], moDC cluster (Sirpα, Cd14, Itgam, Cx3cr1 and Ccr2)[46]; and one pDC cluster (Bst2, Ccr9, Siglech, and Ly6d)[14] (Fig. 5b and Supplementary Data 5). This data allowed the clustering of DCs which participate in CAT according to their specific surface markers (Supplementary Fig. 6a). As shown in Fig. 5b, the previously defined thymic moDC subpopulation shared several markers with both cDCs (Itgax, Itgam, Sirpα, and Irf4) and classical tissue resident macrophages (Lyz2, Mertk, and Mafb). Due to the high mRNA expression of molecules associated with antigen processing and presentation by moDC subpopulation (Supplementary Fig. 6b), we tested their capacity to present mTEC-derived antigens and activate antigen specific T cells. Specifically, thymic CD14[+]moDCs isolated from the Aire-HCO mouse model expressing influenza hemagglutinin (HA) under the control of Aire regulatory sequences[47], were co-cultivated with HA-specific CD4[+] T cell hybridoma cells (A5) carrying a GFP-NFAT reporter[4]. While the result demonstrated that thymic CD14[+]moDCs can efficiently present mTEC-derived antigens to T cells (Supplementary Fig. 6c), it seems that their previous detection was obstructed by using the previously established gating strategy (Supplementary Fig. 4a), by which they are indistinguishable from a conventional Sirpα[+] cDC2 subset.

Next, we determined which of the five defined thymic DC clusters expressed the receptors for TLR9/MyD88-induced chemokines/cytokines from mTECs (Figs. 2b, d, e). The heat map analysis of chemokine receptors identified by ddSEQ analysis revealed that most of these receptors were expressed by the Sirpα[+]CD14[+]moDC cluster (Fig. 5c, left panel). Interestingly, each of the TdTOM[+] DC clusters expressed a specific set of chemokine receptors (Fig. 5c).

Having characterized the CAT system with participating subsets of DCs in Foxn1[Cre]ROSA26[TdTOMATO] mice, we used it as a readout to determine the targeting specificity of TEC-dependent TLR9/MyD88 stimulation on these DC subsets. First, in general, TLR9 intrathymic stimulation of Foxn1[Cre]ROSA26[TdTOMATO] mice boosted the frequency of total TdTOM[+] CD11c[+] DCs (Fig. 5d left graph and Supplementary Fig. 6d) as well as the mean fluorescent intensity (MFI) of TdTOM in these cells, demonstrating their enhanced rate of CAT under stimulatory conditions (Supplementary Fig. 6e). Second, as predicted, the observed increase in CAT was fully attributable to TdTOM[+]Sirpα[+] DCs and not to other DCs populations (Fig. 5d right graph and Supplementary Fig. 6f). Third, and most importantly, the unsupervised flow cytometry tSNE

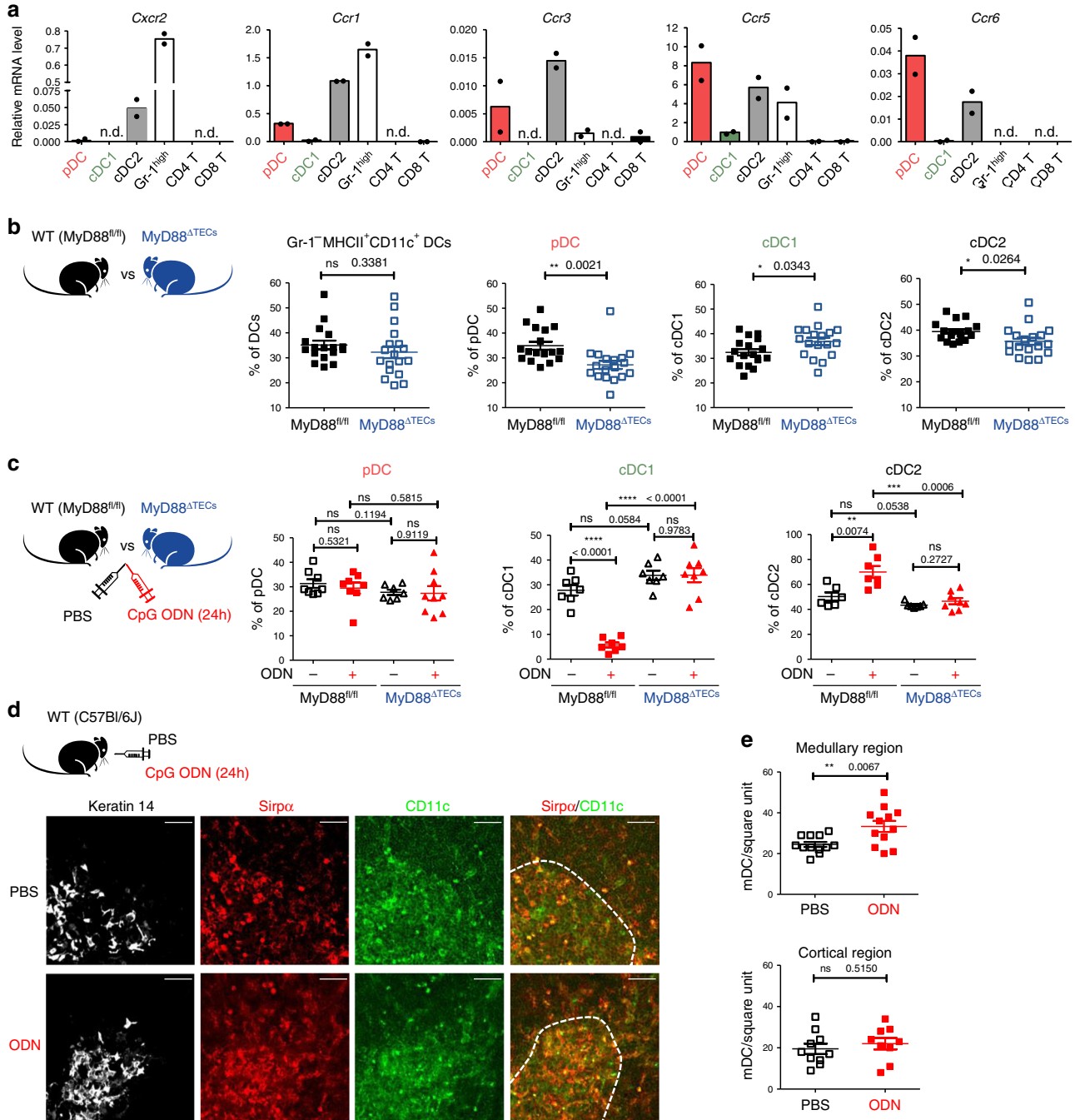

**Fig. 3 TLR/MyD88 signaling in mTECs^high affects the migration of DCs into the thymic medulla. a** qRT-PCR analysis of the relative mRNA expression (normalized to Casc3) of indicated chemokine receptors on FACS sorted populations of thymic DCs; pDC plasmacytoid DCs, cDC1 classical type 1 DC, cDC2 classical type 2 DC, Gr-1^high = neutrophils, CD4 T = CD4+, and CD8 T = CD8+ thymic T cells. Sorting protocol of thymic DC subsets is provided in Supplementary Fig. 4a. T cells were sorted as TCRβ+ and either CD4 or CD8 single positive (n = 2 independent experiments). **b** Comparative flow cytometry analysis of total DCs (Gr-1−CD11c+MHCII+) and different thymic DC subpopulations between MyD88^fl/fl and MyD88^ΔTECs mice enumerated according to gating strategy shown in Supplementary Fig. 4a (mean ± SEM, n = 17 for MyD88^fl/fl and n = 18 for MyD88^ΔTECs mice). **c** Flow cytometry analysis of different thymic DC populations (gated as in Supplementary Fig. 4a) isolated from CpG ODN or PBS intrathymically stimulated MyD88^fl/fl or MyD88^ΔTECs mice (mean ± SEM, pDC graph: n = 7 for ODN−MyD88^ΔTECs and n = 8 for other three displayed items; cDC1 graph: n = 7 for MyD88^fl/fl and ODN−MyD88^ΔTECs and n = 8 for ODN+ MyD88^ΔTECs mice; cDC2 graph: n = 6 for ODN−MyD88^fl/fl, n = 7 for ODN+MyD88^fl/fl and ODN−MyD88^ΔTECs and n = 8 for ODN+MyD88^ΔTECs mice). Statistical analysis in **b**, **c** was performed by unpaired, two-tailed Student's t-test, p ≤ 0.05 = *, p ≤ 0.01 = **, p ≤ 0.001***, p < 0.0001 = ****, ns not significant. **d** Microscopic examinations of thymic sections isolated from CpG ODN or PBS intrathymically stimulated WT mice. Cryosections were stained with keratin 14 (white), Sirpα (red), and CD11c (green). Scale bar represents 50 μm. The white dashed line demarks keratin 14-rich medulla. **e** Quantification of CD11c+Sirpα+ cells in the medullar or cortical region of the cryosections shown in d (mean ± SEM, n = 12 counted square unites per medullary region; n = 10 and n = 9 counted square unites per PBS- and ODN-treated cortical region, respectively. Data are derived from three independent experiments). Statistical analysis was performed by unpaired, two-tailed Student's t-test, p ≤ 0.01 = **, ns not significant.

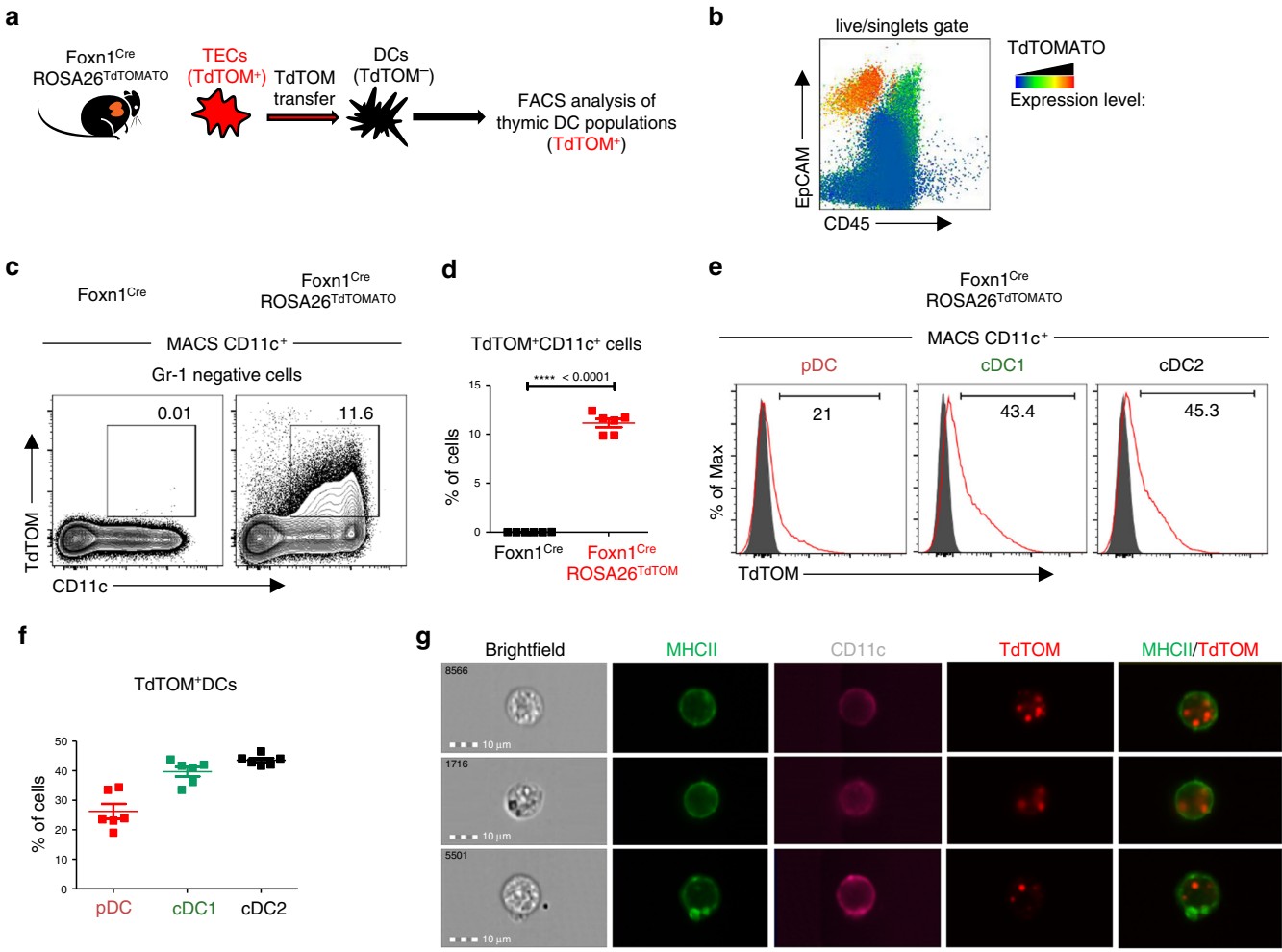

**Fig. 4 Foxn1^CreROSA26^TdTOMATO as a model of thymic cooperative antigen transfer. a** Experimental design. **b** Flow cytometry heat-map analysis showing the intensity of TdTOM fluorescence among MACS TCRβ-depleted cells from the thymus of the Foxn1^CreROSA26^TdTOMATO mouse. **c** Representative flow cytometry plots comparing the frequency of TdTOM^+CD11c^+ cells in the thymic MACS-enriched CD11c^+ cells between the WT (Foxn1^Cre) and Foxn1^CreROSA26^TdTOM mouse. Cells were pre-gated as live, singlets, and Gr-1^-. **d** Quantification of TdTOM^+CD11c^+ cells from c (mean ± SEM, n = 6 mice). Statistical analysis was performed by unpaired, two-tailed Student's t-test, p < 0.0001 = ****. **e** Representative flow cytometry histograms showing the frequency of TdTOM^+ cells among pDC, cDC1, and cDC2 (gated as in Supplementary Fig. 4a). Gray histograms = Foxn1^Cre (control) mice, red histograms = Foxn1^CreROSA26^TdTOM mice. **f** Quantification of frequencies of TdTOM^+ DCs among the indicated DC subsets (mean ± SEM, n = 6 mice). **g** Representative images from the Imagestream analysis showing intracellular localization of transferred TdTOM in MHCII^+CD11c^+ DCs from the thymus of Foxn1^CreROSA26^TdTOMATO (n = 400 measured cells).

analysis of the main DC subsets defined by markers revealed by ddSEQ analysis showed that the increase of TdTOM^+ DCs was mostly due to the specific enrichment of CD14^+moDCs (Figs. 5e, f), which also co-express chemokine receptors for ligands induced by TLR9/MyD88 signaling in mTECs (Figs. 2b, e and 5c). Concomitantly, we observed a decrease in Mgl2^+ cDC2, Xcr1^+ cDC1b, and B220^+ pDC (Fig. 5e, f and Supplementary Fig. 6g). Importantly, and further confirming the need of MyD88 signaling for its recruitment, the decreased frequency of total Sirpα^+ DCs in the thymus of non-manipulated MyD88^ΔTECs mice (Fig. 3b) was shown to be accounted specifically by the diminishment of the CD14^+moDC subset (Fig. 5g).

To find which of the chemokines described (in Fig. 2b, e) were responsible for CD14^+moDC migration to the thymus, we crossed Cxcr2^fl/fl mice with the pan-hematopoietic driver Vav1^Cre to abrogate the signaling of its cognate ligands Cxcl1, 2, 3, and 5 that were among the most upregulated genes in mTECs after TLR9 stimulation. We observed no changes in the recruitment of CD14^+moDC after TLR9 stimulation between Cxcr2^fl/flVav1^Cre

and WT mice (Supplementary Fig. 6h). This suggests, that together with ligands of Ccr2, (i.e. Ccl2, 7, 8, and 12)[19], the ligands of Ccr1, Ccr3 or Ccr5, or their combinations[36], regulate the entry of CD14^+moDC into the thymic medulla.

Together, TLR9/MyD88-dependent chemokine signaling in mTECs specifically targets the recruitment and subsequent CAT from the mTECs to Sirpα^+CD14^+moDC subpopulation which exhibits a tangible capacity for antigen presentation.

**TLR9/MyD88 signaling in mTECs affects Treg development.** Previous studies have suggested that the development of thymic Tregs is dependent on antigen presentation by both mTECs and DCs[6,17,47]. Specifically, antigen presentation by Sirpα^+DCs[17] and/or alternatively by CD8α^+cDC1[6,10] was implied in the development of organ-specific Tregs. It has been also suggested that the increased ratio of Sirpα^+DCs to CD8α^+cDC1 leads to an enhanced production of thymic CD25^+Foxp3^+ Tregs[17,20]. Since a decreased frequency of Sirpα^+DCs (Fig. 3b), specifically CD14^+moDCs (Fig. 5g) was observed in the thymus of

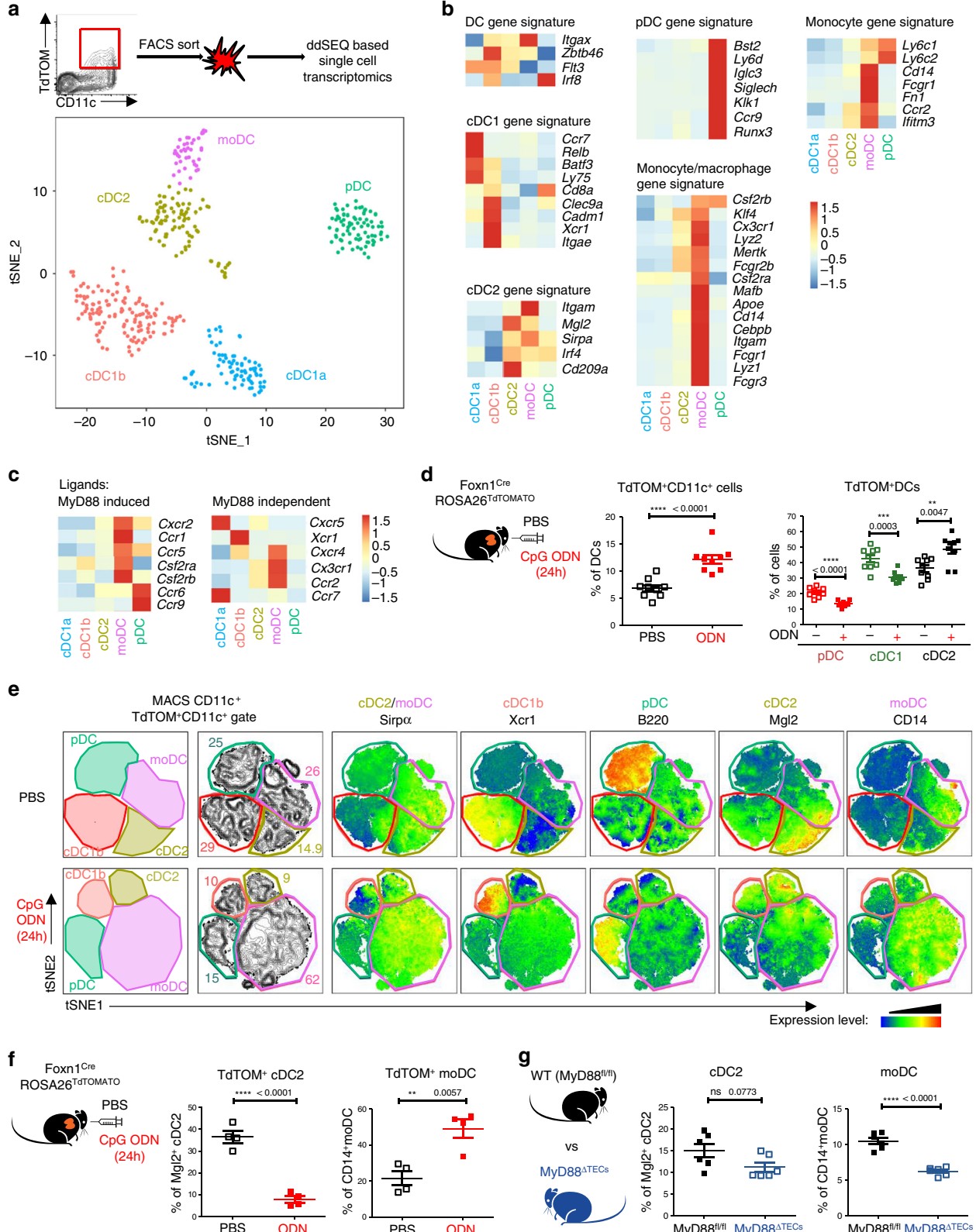

MyD88$^{\Delta TECs}$ mice, we tested whether these effects would impact the development of the major thymocyte populations and Tregs. While the DN (CD8$^-$CD4$^-$), DP (CD8$^+$CD4$^+$), and CD8$^+$ T cells frequencies were comparable between MyD88$^{\Delta TEC}$ and WT mice, CD4$^+$ T cells, and more specifically CD25$^+$Foxp3$^+$ Tregs were significantly reduced in 4-week-old MyD88$^{\Delta TECs}$ mice (Fig. 6a–c and Supplementary Fig. 7a). Since it has been reported that in 4

week-old-mice nearly one half of CD25$^+$Foxp3$^+$ thymic cells consist of mature recirculating Tregs[48,49], we used CD73 protein staining to determine if Tregs reduced in MyD88$^{\Delta TECs}$ mice were newly generated (CD73$^-$) or recirculating (CD73$^+$)[50]. As shown in Fig. 6d, e, the abrogation of MyD88 signaling in mTECs affected mainly the generation of CD25$^+$Foxp3$^+$ thymic Tregs and not their recirculation. On the other hand, the CD25$^+$Foxp3$^+$ thymic Tregs

**Fig. 5 TLR/MyD88 signaling increases cooperative antigen transfer between TECs and the CD14+moDC subpopulation. a** Two-dimensional tSNE plot from ddSEQ single-cell RNA-sequencing from FACS sorted Gr-1−CD11c+TdTOM+ DCs from the thymus of Foxn1CreROSA26TdTOMATO mice. The color code represents different cell clusters based on the mRNA expression profile of each cell. **b** Heat-map analysis of the expression of signature genes determining each subset defined in **a**. **c** Heat-map analysis of the expression of chemokine receptors by DC subsets defined in **a**. **d** Quantification of TdTOM+CD11c+ DC subsets (defined as in Supplementary Fig. 4a) in CpG ODN or PBS intrathymically stimulated Foxn1CreROSA26TdTOMATO mice (representative flow cytometry plots are shown in Supplementary Fig. 6d, f) (mean ± SEM, $n = 9$ mice). Statistical analysis was performed using unpaired, two-tailed Student's $t$-test, $p \leq 0.01 = $**, $p \leq 0.001$***, $p < 0.0001$****. **e** Representative flow cytometry tSNE analysis of TdTOM+CD11c+cell population in PBS or CpG ODN intrathymically stimulated Foxn1CreROSA26TdTOMATO mice. tSNE analysis was performed using FlowJO software, based on the FSC-A, SSC-A, CD11c, MHCII, Sirpα, Xcr1, B220, Mgl2 and CD14 markers ($n = 2$ independent experiments). **f** Quantification of frequencies of TdTOM+CD14+ moDC or TdTOM+Mgl2+cDC2 from CpG ODN or PBS intrathymically stimulated Foxn1CreROSA26TdTOMATO mice (representative flow cytometry plots are shown in Supplementary Fig 6g) (mean ± SEM, $n = 4$ mice). **g** Flow cytometry analysis comparing the frequency of cDC2 (Sirpα+Mgl2+) and moDC (Sirpα+CD14+) between MyD88fl/fl and MyD88ΔTECs mice (mean ± SEM, $n = 6$ mice). Total Sirpα+ DC population was gated as shown in Supplementary Fig 4a. Statistical analysis in f and g was performed by unpaired, two-tailed Student's $t$-test, $p \leq 0.01 = $**, $p < 0.0001$****, ns not significant.

were not reduced in newborn MyD88ΔTECs (Supplementary Fig. 7b) or GF mice (Supplementary Fig. 7c) when compared to their WT SPF littermates. This, in association with unchanged chemokine expression in mTECs from GF mice, (Supplementary Fig. 2b) further strengthens the notion that the ligands that regulate the mTEC-mediated MyD88-dependent cellularity of Tregs is not likely of exogenous origin.

To further explore the MyD88-dependent regulation of Tregs generation, we tested our prediction that TLR9/MyD88 stimulation of mTECs would lead to the opposite effect, i.e. boosted number of Tregs. Indeed, seven days after intrathymic injection of CpG ODN, we observed a significant increase in the frequency and total number of CD25+Foxp3+ thymic Tregs (Fig. 6f and Supplementary Fig. 7d–f). Importantly, this increase was completely dependent on TEC-intrinsic MyD88 signaling (Fig. 6f). Compared to the decreased numbers in CD73− Tregs in MyD88ΔTEC, intrathymic injection of CpG ODN led to increased numbers of not only CD73− newly generated Tregs but also recirculating CD73+ Tregs (Fig. 6g and Supplementary Fig. 7g). This suggests that there are other mTEC-dependent mechanisms which after CpG ODN stimulation can affect the recirculation of Tregs into the thymus. One outstanding question related to the results from the above experiments (Figs. 3c and 5d–f and Supplementary Fig. 6c) is whether the increased generation of thymic CD25+Foxp3+CD73− thymic Tregs is dependent on the antigen presenting capacity of DCs. To resolve this query, we intrathymically injected CpG ODN into H2-Ab1fl/flItgaxCre (H2-Ab1ΔDCs) mice, where antigen presentation by DCs has been abrogated. As demonstrated in Fig. 6h, i and Supplementary Fig. 7h, the presentation of antigen by DCs is essential for the increase in numbers of newly generated CD73−CD25+Foxp3+ thymic Tregs after TLR9 stimulation.

Next, we tested the physiological consequences of the decrease in production of Tregs in MyD88ΔTECs mice. We took advantage of a T cell induced colitis model, where the adoptive transfer of naïve, Treg depleted CD4+ T cells into Rag1-deficient mice induces severe colitis[51]. In this experimental setup, and as illustrated in Fig. 7a, the i.p. injection of the CD4+ T cell population isolated from peripheral lymph nodes of either MyD88ΔTEC or MyD88fl/fl mice was compared to colitis-inducing transfer of CD4+CD45RBhighCD25− cells isolated from WT mice.

Strikingly, mice that received CD4+ T cells from MyD88ΔTECs began to lose weight ~4 weeks after adoptive transfer, behaving identically to the positive control. In contrast, mice that received CD4+ T cells from WT mice continuously gained weight over time (Fig. 7b). The clinical signs of colitis in mice receiving CD4+ T cells from MyD88ΔTEC and in the positive controls were manifested by the presence of inflammatory infiltrates in the colon lamina propria, increased bowel wall thickness, presence of abscesses in colon tissue (Fig. 7c), increased spleen weight (Supplementary

Fig. 8a, b), and a higher colon weight/length ratio (Fig. 7d and Supplementary Fig. 8a). To confirm the persistence of the transferred T cell population, we also analyzed Tregs frequencies in all conditions. We found that both positive controls and mice that received CD4+ T cells from MyD88ΔTECs had severely diminished Tregs compared to WT controls (Fig. 7e). The very similar phenotype of mice that received CD4+ T cells from MyD88ΔTECs and those which received CD4+CD45RBhighCD25− suggested, that Tregs in MyD88ΔTECs were not only reduced in numbers but also functionally altered. Along with the decreased expression of CD25 (Fig. 7f), Tregs from MyD88ΔTECs mice showed a significantly reduced capacity to suppress the proliferation of OVA-specific OT-II T cells in vitro (Fig. 7g, h) and prevent the early onset of diabetes caused by activated KLGR1+ OT-I T cells in a RIP-OVA dependent autoimmune mouse model[52] (Supplementary Fig. 8c–e).

Taken together, these results demonstrate that TLR/MyD88 signaling in TECs affects the development of thymic CD25+Foxp3+ Tregs. Specifically, in mice with MyD88-deficient TECs, the frequency and functionality of thymic CD25+Foxp3+ Tregs was decreased and unable to prevent T cell induced colitis.

## Discussion

Present study lends a support for the role of TLR signaling in the mechanism of central tolerance. First, we found that mTECshigh express TLRs, including TLR9, whose signaling is functionally wired to the expression of chemokines and genes associated with their post-Aire development. Second, the receptors for these chemokines are predominantly expressed by the Sirpα+ thymic population of CD14+moDCs whose enrichment in the thymus and subsequent CAT is positively regulated by mTEC-intrinsic TLR/MyD88 signaling. Third, TLR/MyD88 signaling in mTECs is important for the proper development of thymic CD73−CD25+Foxp3+ Tregs since its abrogation resulted in a decreased number and the functionality of Tregs, associated with pathological effects in the mouse model of colitis.

The importance of TLR/MyD88 signaling in Aire-dependent autoimmunity was suggested in experiments conducted with MyD88−/−Aire−/− double-knockout mice. These mice develop more severe symptoms of autoimmunity than Aire−/− single KO animals indicating the positive regulatory role of MyD88 signals in tolerance induction. Strikingly, neither the enhancement of MyD88 signals by an i.p. injection of TLR ligands, nor their diminishment in mice from GF conditions altered the severity of Aire-dependent autoimmunity[53]. Our data advocates for a scenario in which the worsening of autoimmunity in MyD88−/−Aire−/− mice could be caused by the lack of MyD88 signaling in mTECshigh, downregulation of their chemokines needed to

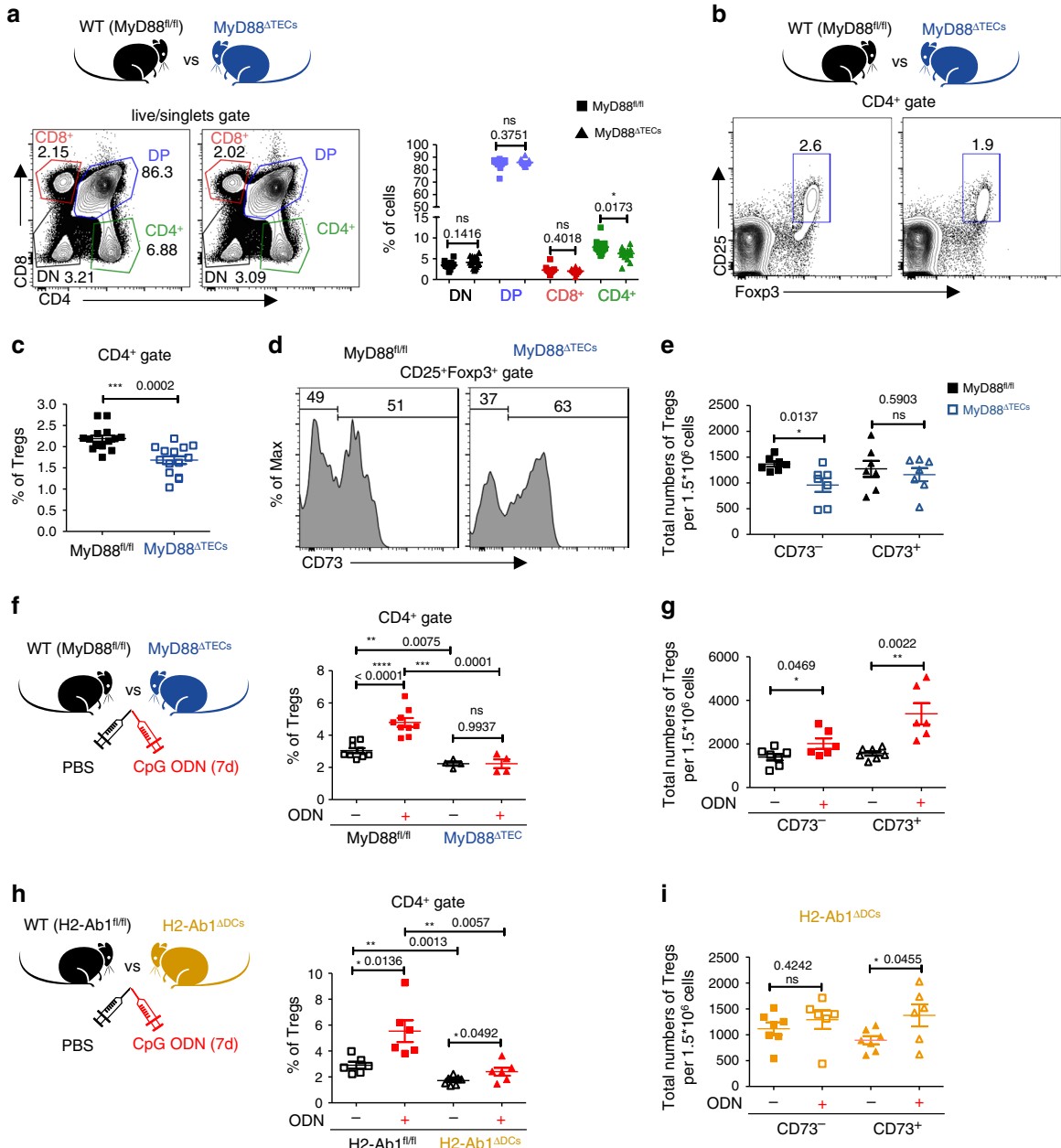

**Fig. 6 Development of thymic Tregs is impaired in MyD88$^{\Delta TECs}$ mice. a** Representative flow cytometry plots (left plot) and their quantification (right plot) comparing the frequencies of main thymic T cell populations between MyD88$^{fl/fl}$ and MyD88$^{\Delta TECs}$ mice (mean ± SEM, $n = 14$ mice). **b** Representative flow cytometry plots comparing the frequencies of CD4$^+$CD25$^+$Foxp3$^+$ thymic Tregs between MyD88$^{fl/fl}$ and MyD88$^{\Delta TECs}$ mice. **c** Quantification of frequencies from b (mean ± SEM, $n = 14$ mice). **d** Representative flow cytometry histograms showing the expression of CD73 by CD4$^+$ CD25$^+$Foxp3$^+$ thymic Tregs (gated as in b). **e** Quantification of the total numbers of CD73$^-$ and CD73$^+$ thymic Tregs from d (mean ± SEM, $n = 7$ mice). **f** Quantification of the frequencies of thymic Tregs from CpG ODN or PBS intrathymically stimulated (7 days) MyD88$^{fl/fl}$ or MyD88$^{\Delta TECs}$ mice (mean ± SEM, $n = 4$ for MyD88$^{\Delta TECs}$ and $n = 9$ for MyD88$^{fl/fl}$ mice). **g** Quantification of the total numbers of CD73$^-$ and CD73$^+$ thymic Tregs from CpG ODN or PBS intrathymically stimulated (7 days) WT (C57Bl/6J) mice (mean ± SEM, $n = 6$ for ODN$^+$ and $n = 7$ for ODN$^-$ mice). **h** Quantification of frequencies of thymic Tregs from CpG ODN or PBS intrathymically stimulated (7 days) H2-Ab1$^{fl/fl}$ or H2-Ab1$^{fl/fl}$Itgax$^{Cre}$ (H2-Ab1$^{\Delta DCs}$) mice (mean ± SEM, $n = 6$ for H2-Ab1$^{fl/fl}$ and ODN$^+$ H2-Ab1$^{\Delta DCs}$ and $n = 7$ for ODN$^-$ H2-Ab1$^{\Delta DCs}$ mice). **i** Quantification of the total numbers of CD73$^-$ and CD73$^+$ thymic Tregs from CpG ODN or PBS intrathymically stimulated (7 days) H2-Ab1$^{\Delta DCs}$ mice (mean ± SEM, $n = 6$ for ODN$^+$ and $n = 7$ for ODN$^-$ mice). Statistical analysis in **a**, c, **e–i** was performed by unpaired, two-tailed Student's $t$-test, $p \leq 0.05 = *$, $p \leq 0.01 = **$, $p \leq 0.001***$ $p < 0.0001****$, ns not significant.

recruit CD14$^+$moDCs and, consequently, suboptimal production of thymic Tregs. Consistent with the previous report[53], we confirmed that the extrathymically enhanced (i.p. CpG ODN) or the lack of bacterially-derived MyD88 signals (GF mice) had no effect on the expression level of these chemokines and cytokines in WT mice. This was further corroborated by the fact that GF mice displayed normal numbers of Tregs[50] (Supplementary Fig. 7c).

This data demonstrates that the ligand triggering TLR9/ MyD88 signaling in mTECs$^{high}$ is likely of endogenous thymic-derived origin.

Since MyD88 also conveys signals from the receptors of IL-1 family cytokines (IL-1β, IL-18, IL-33)[38], we tested in vitro whether their signaling in mTECs$^{high}$ could trigger chemokine responses similar to those observed upon TLR9 stimulation. Of

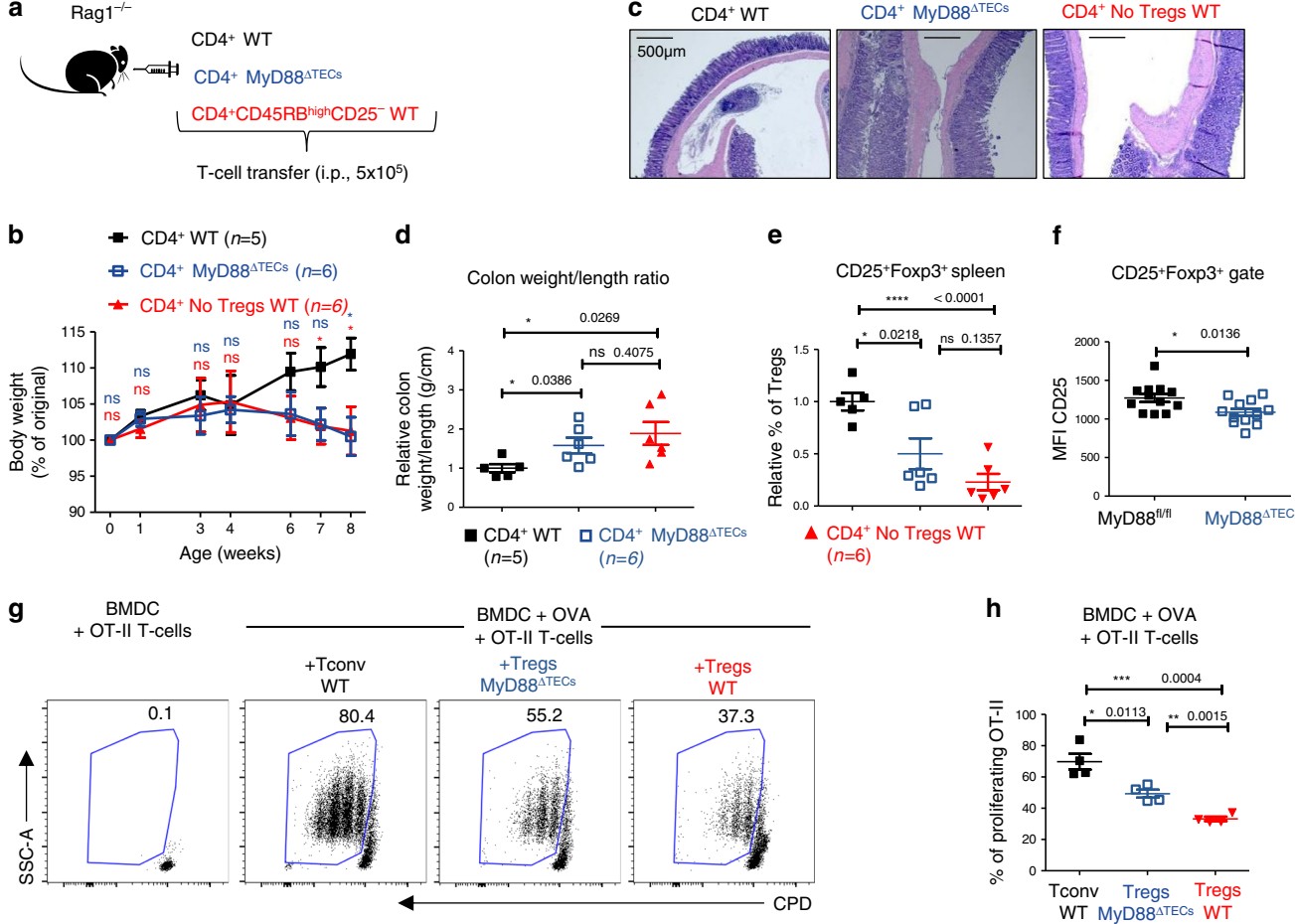

**Fig. 7 Tregs from MyD88$^{\Delta TECs}$ mice have reduced suppressive capacity and failed to prevent the T cell induced colitis. a** Experimental design of induced colitis. **b** Relative quantification of mice weight normalized to its value on day 0 (100% of original weight) after T cell transfer over the time-course of the colitis experiment (mean ± SEM, $n = 5$–6 mice) Statistical analysis was performed by unpaired, two-tailed Student's $t$-test comparing the relative weight of WT CD4$^+$ with MyD88$^{\Delta TECs}$ CD4$^+$ transferred mice (blue) or with WT CD4$^+$CD45RB$^{high}$CD25$^-$ transferred mice (red), $p \leq 0.05 = *$, ns not significant. **c** Representative H&E-stained slides of colon sections performed 8 weeks after T cell transfer. Scale bar represents 500 µm ($n = 5$ for CD4$^+$ WT and $n = 6$ for CD4$^+$MyD88 $^{\Delta TECs}$ and CD4$^+$ No Tregs WT mice). **d** Relative quantification (normalized to average of control mice from each experiment) of colon weight/length ratio of T cell induced colitis experimental mice (mean ± SEM, $n = 5$–6 mice). **e** Relative quantification of the frequencies (normalized to average of control mice from each experiment) of CD4$^+$CD25$^+$Foxp3$^+$ Tregs isolated from the spleens of experimental mice 8 weeks after T cell transfer (mean ± SEM, $n = 5$–6 mice). **f** Quantification of the Means fluorescent intensity (MFI) of CD25 protein expression in CD25$^+$Foxp3$^+$ Tregs (gated as in Fig. 6b) in MyD88$^{fl/fl}$ and MyD88$^{\Delta TECs}$ mice (mean ± SEM, $n = 12$ mice) Statistical analysis in **b**, **d**-**f** was performed by unpaired, two-tailed Student's $t$-test, $p \leq 0.05 = *$, $p < 0.0001****$, ns not significant. **g** Representative flow cytometry plots showing the frequency of proliferating OT-II T cells, co-cultivated with OVA pulsed BMDC and CD4$^+$CD25$^+$ Tregs cells (alternatively with CD4$^+$CD25$^-$ Tconv cells, black) isolated from LNs of MyD88$^{fl/fl}$ (WT control, red) or MyD88$^{\Delta TECs}$ (blue) for 72 h. **h** Quantification of frequencies of proliferating OT-II Tcells form h (mean ± SEM, $n = 4$ wells from two independent experiments).

this trio of cytokines, only IL-1β exhibited this capacity. This indicates that IL-1β could act as a co-regulator of chemokines and cytokine expression in mTECs$^{high}$. However, two observations suggest that TLR9/MyD88 signaling axis can act independently of IL-1β: (i) a direct, in vitro, stimulatory capacity of CpG ODN induces chemokine expression in sorted mTECs$^{high}$; and (ii) both in vivo intrathymic stimulation of TLR9/MyD88 signaling axis as well as its downregulation in MyD88$^{\Delta TECs}$ cells impacts the recruitment of the very same subsets of CD14$^+$moDCs.

It has been postulated that Aire$^+$ mTECs further differentiate into post-Aire cells, which downregulate the expression of MHCII and Aire, upregulate a set of genes, such as keratins (Krt1, 10, 77) or involucrin and form Hassall's corpuscules[40,41,54]. However, the regulatory mechanism(s) guiding this differentiation process remains poorly understood[55]. Our transcriptomic results are consistent with the idea that TLR/MyD88 signaling establishes an

expression profile that is associated with the differentiation of mTECs$^{high}$ into post-Aire mTECs. Notably, TLR9 stimulation not only increased the number of Involucrin$^+$ post-Aire mTECs (Supplementary Fig. 3e, f), but also lead to the upregulation of cytokines and chemokines (*Il1f6, Lcn2, Cxcl3,* and *Cxcl5*) associated with Hassall's corpuscules[42] which attract CD14$^+$moDCs. Together with the fact that they serve as a reservoir of a large amount of Aire-dependent TRAs, post-Aire mTECs could hold central position in the mechanism of transfer of mTEC-derived antigens to thymic DCs.

As described above, TLR/MyD88 signaling in mTECs$^{high}$ drive the expression of chemokines which act on an overlapping set of receptors[32] predominantly expressed by CD14$^+$moDCs (Cxcr2, Ccr1, Ccr3, Ccr5) and pDCs (Ccr5, Ccr6, and Ccr9). A correlative nature between the frequency of CD14$^+$moDC in the thymus of MyD88$^{\Delta TECs}$ and of WT stimulated with CpG, underpins the

importance of these chemokines in controlling the migration of these cells into the thymic medulla. However, the deletion of Cxcr2 on hematopoietic cells, the common receptor for Cxcl1, Cxcl2, Cxcl3 and Cxcl5, did not yield any changes in the enrichment of CD14+moDC in the thymus (Supplementary Fig. 6h). This observation, in conjunction with previous reports[18,56], allows one to predict that while the ligands of Ccr3 and/or Ccr5 (Ccl3, Ccl4, Ccl5 or Ccl24) likely regulate the entry of CD14+moDC into the thymus[19], Cxcl-chemokines may regulate the positioning of these cells in close proximity of post-Aire mTECs. Interestingly, with the decreased frequency of CD14+ moDCs in the thymus of MyD88ΔTEC, pDCs were similarly diminished. However, in contrast to CD14+moDCs, the number of pDCs did not increase after TLR9 intrathymic stimulation. This is consistent with the fact that the migration of pDCs to the thymus is driven by Ccl25 (ligand of Ccr9 receptor)[14], the expression of which was diminished in MyD88ΔTEC but was not upregulated in WT mTECs after TLR9 stimulation.

It has been previously documented that specific subtypes of thymic DCs vary in their capacity to acquire antigens from TECs. Notably, while the transfer of MHC molecules from TECs to CD8α+ cDC1 and Sirpα+DCs occurred at the same efficiency[16], the transfer of intracellular GFP was restricted mainly to CD8α+cDC1[10]. In comparison, our data shows that cytoplasmic TdTOM from Foxn1CreROSA26TdTOMATO could to certain extent, be transferred to all major subtypes of thymic DCs. This may be explained by the robustness of the Foxn1Cre-dependent system where, compared to Aire-GFP model, the production of TdTOM is not restricted only to Aire-expressing mTECs but to the entire thymic TEC population. Importantly, since the CAT of TdTOM after CpG ODN intrathymic injection is increasingly targeted to CD14+moDC subpopulation, the efficiency of CAT correlates not only with the broadness of antigen expression but also with the frequency of a given DC subtype in the medulla. On the other hand, since TECs constitute a relatively rare cell population of thymic cells[57], the amount of antigen, which can be potentially transferred to DCs, is fairly limited. This could explain the fact that even when the entire population of thymic pDCs is not affected by intrathymic TLR9 stimulation, the frequency of TdTOM+ pDCs is significantly decreased, due to the increased competition for TdTOM uptake by CD14+moDCs.

It has become clear that developing thymocytes encounter self-antigens presented by various types of thymic APC, including mTECs[47], B-cells[58], pDCs[14], and cDCs[11,59]. Although the generation of thymic Tregs was shown to be dependent on antigen presentation by both mTECs and DCs[4,47], thymic cDCs seem to be particularly important for this process[6,17,60]. Along with self-antigen presentation, thymic cDCs express high levels of co-stimulatory molecules CD80/86 as well as CD70 which play a crucial role in promoting thymic Treg development[61,62]. Among cDCs, Sirpα+DCs are the most efficient in supporting Treg generation[17,20,63]. In this context, our data demonstrates that the development of thymic CD25+Foxp3+ Tregs is boosted by TLR/MyD88 signaling in TECs, which produce a chemokine gradient driving the migration of CD14+moDCs into the thymus. We also found that mTEC-intrinsic TLR9/MyD88 signaling increased the cell ratio of Sirpα+DCs to Xcr1+cDC1, which correlated with an increased production of thymic Tregs. These findings accurately recapitulate the thymic phenotype of Ccr7−/− mice where the increased ratio of Sirpα+DCs to cDC1 correlated with the increased generation of thymic Tregs[20]. This data, together with the fact that abrogation of MHCII-antigen presentation specifically in DCs, resulted in a reduced number of thymic Tregs in unstimulated[17] as well as in CpG stimulated thymus (Fig. 6h), suggest that TLR/MyD88-dependent generation of thymic Tregs is mediated by antigen-presentation by DCs.

Our results also show that TLR/MyD88 signalling in mTECs drives the recirculation of mature CD73+CD25+Foxp3+ Tregs into the thymus. Compared to newly generated CD73− Tregs, their increased number in the TLR9 stimulated thymus was not dependent on MHCII presentation by DCs. Together, with the fact that recirculation of CD73+ Tregs was not abrogated in MyD88ΔTECs mice, suggests that Ccl20, the ligand for Ccr6, which is highly expressed by recirculating Tregs[64] regulates the increased recirculation of CD73+CD25+Foxp3+ Tregs into the thymus after TLR9 intrathymic stimulation (Figs. 2d and e).

Altogether, our model proposes that TLR/MyD88 signaling in mTECs regulates the generation of Tregs. The mechanism involves TLR-induced chemokine production and subsequent chemotactic recruitment of CD14+moDC to the thymic medulla, which predicates the developmental output of Tregs. Although this study explores only TLR9 signaling in mTECs, questions surrounding the nature of potential thymic-derived endogenous ligands for TLR/MyD88 signals in mTECs remains enigmatic and warrant further study.

## Methods

**Mice**. A majority of the mice used in this study were of C57BL/6J genetic background and housed in the animal facility at the Institute of Molecular Genetics of the ASCR v.v.i. under SPF conditions. Mice were fed with irradiated standard rodent high energy breeding diet (Altromin 1314 IRR) and given reverse osmosis filtered and chlorinated water ad libitum. Light were adjusted to a 12 h/12 h light/dark cycle; temperature and relative humidity were maintained at 22 ± 1°C and 55 ± 5%, respectively. Experimental protocols were approved by the ethical committee of the Institute of Molecular Genetics and by the ethical committee of the Czech Academy of Science. Aire−/− (B6.129S2-Airetm1.1Doi/J, stock# 004743)[2], Foxn1Cre (B6(Cg)-Foxn1tm3(cre)Nrm/J, stock# 018448)[29], MyD88fl/fl (B6.129P2(SJL)-Myd88tm1.1Defr/J, stock# 008888), MyD88−/− (B6.129P2(SJL)-Myd88tm1.1Defr/J,, stock# 009088)[30], Rag1−/− (B6.129S7-Rag1tm1Mom/J, stock# 002216)[65], Ly5.1 (B6.SJL-PtprcaPepcb/BoyJ, stock# 002014)[66], Cxcr2fl/fl (C57BL/6-Cxcr2tm1Rmra/J, stock# 024638)[67], H2-Ab1fl/fl (B6.129×1-H2-Ab1tm1Koni/J, stock# 013181)[68], and ItgaxCre (B6.Cg-Tg(Itgax-cre)1-1Reiz/J, stock# 008068)[69] mice were purchased from Jackson Laboratories. Rosa26TdTOMATO (B6;129S6-Gt(ROSA)26Sortm14(CAG-tdTomato)Hze/J, stock# 007908)[70] and Vav1Cre (B6.Cg-Commd10Tg(Vav1-icre)A2Kio/J, stock# 008610)[71] were kindly provided by V. Kořínek (Institute of Molecular Genetics of the ASCR, Prague, Czech Republic). Aire-HCO (Balb/c)[4] were provided by L. Klein. Cd3e−/−[72], RIP-OVA[73], OT-I+Rag2−/−[74] (all C57BL/6J) were provided by O. Štěpánek. OT-II (B6.Cg-Tg(TcraTcrb)425Cbn/J, stock# 004194)[75] mice were kindly provided by T. Brdička (Institute of Molecular Genetics of the ASCR, Prague, Czech Republic). C57BL/6J GF and control C57BL/6J SPF mice were kindly provided by M. Schwarzer (Institute of Microbiology of the ASCR, Nový Hrádek, Czech Republic. Both GF and control SPF mice were subject to the SSNIFF V1124-300 diet. Thymic cell populations were isolated from 3–6-week-old mice with the exception of newborn mice (4 days old) used in Supplementary Fig. 7b. For the purpose of BM chimera experiments, 5–6-week-old mice were irradiated and analysed between 11 and 13 weeks of age. Comparative analysis used age-matched cohorts regardless of sex and caging. Where possible, littermates were used as the controls. For the purpose of tissue isolation, mice were euthanized by cervical dislocation.

**Tissue preparation and cell isolation**. Thymic antigen presenting cells, TECs and DCs, were isolated as follows. Thymus was minced into small pieces and treated with Dispase II (Gibco), dissolved in RPMI at concentration 0.1 mg ml−1. Tissue was homogenized by pipetting and after 10 min of incubation (37°C), the supernatant was collected and the reaction was stopped by adding 3% FSC and 2 nM EDTA. The process was repeated until all thymic fragments were digested. For detailed description see[76]. For thymic epithelial cells isolation, the whole thymic cell suspension was depleted of CD45+ cells by CD45 microbeads staining (Miltenyi biotec). Thymic dendritic cells were isolated using MACS enrichment for CD11c+ cells through staining with biotinylated CD11c antibody, followed by Ultrapure Anti-Biotin microbeads staining (Miltenyi biotec). For isolation of T cell, thymus, peripheral lymph nodes (pLN), mesenteric lymph nodes (mLN) or spleen were mechanically mashed through 40 μm Cell strainer (Biologix) and cell suspensions were passed through 50 μm filters (Sysmex). The resulting cell suspension was spun down (4 °C, 400 g, 10 min) and erythrocytes were removed using ACK lysis buffer.

**Flow cytometry analysis and cell sorting**. Flow cytometry (FACS) analysis and cell sorting were performed using BD LSR II and BD Influx (BD Bioscience) cytometers, respectively. For surface staining, cells were incubated for 20–30 min at 4 °C with the indicated fluorochrome- or biotin-conjugated antibodies. Where necessary, cells were further incubated with streptavidin conjugates for 15 min.

Dead cells were excluded using Hoechst 33258 (Sigma) or viability dye eFlour 450 or 506 (eBioscience). For the intracellular staining of Aire and Foxp3, the cells were first stained for the targeted surface molecules, fixed, and permeabilized for 30 min at room temperature (RT) using the Foxp3/Transcription Factor Staining Buffer Set (eBioscience), then stained for 30 min at RT with fluorochrome-conjugated antibodies. FlowJO V10 software (Treestar) and BD FACSDiva™ Software v6.0 for BD™ LSR II (with HTS Option) was used for FACS data analysis including tSNE analysis shown in Fig. 5e. A complete inventory of staining reagents is listed in Supplementary Data 6.

**Imaging flow cytometry.** Imaging flow cytometry was performed at the Center for Advanced Preclinical Imaging (CAPI) with the use of AMNIS ImageStream X MkII (AMNIS). DCs isolated from Foxn1$^{Cre}$ROSA26$^{TdTOMATO}$ mice were stained for the surface markers MHCII and CD11c. Dead cells were excluded by Hoechst 33258 staining and bright field analysis. Cells were recorded using 40x magnification. Data was analyzed with Ideas 6.1 software (AMNIS). A complete list of staining reagents can be found in Supplementary Data 6.

**In vitro TLRs and cytokines stimulation assays.** mTECs$^{high}$ were gated as EpCAM$^+$CD11c$^-$Ly51$^-$MHCII$^{high}$CD80$^{high}$ and sorted into RPMI media (Sigma) containing 10% FSC and 1% Penicillin/Streptomycin (Gibco). Cells were then cultured in a 96-flat-well plate in 200 μL of 10% FSC RPMI with Penicillin/ Streptomycin in the presence of Endotoxin-free TLR ligands (InvivoGen) or recombinant mouse cytokines: TLR9 ligand-CpG ODN (ODN 1826) (5 μM), TLR4 ligand-LPS (1 μg/ml), Il-1β (10 ng/ml), Il-33 (10 ng/ml) (both ImmunoTools) and Il-18 (10 ng/ml) (Biolegend). After 24 h, the supernatant was removed and the cells were resuspended in RNA-lysis buffer. Subsequently, RNA isolation was performed.

**In vivo TLR stimulation.** For intrathymic injections, mice were anesthetized by i.p. injection of Zoletil (Tiletamine (50 mg/ml) and Zolazepam (50 mg/ml), Virbac) dissolved in PBS at a dose of 50 mg/kg and 10–20 μl of 500 μM CpG ODN (ODN 1826, InvivoGen) or PBS was injected using an insulin syringe (29G) directly into the first intercostal space from the manubrium ~2 mm left of the sternum and 4 mm in depth. The angle of injection was from 25 to 30° relative to the sternum[77]. For systemic TLR9 stimulation, mice were injected by CpG ODN (ODN 1826, InvivoGen) (500 μM) or PBS at day 0 and day 1 into the peritoneum. Mice were then maintained under SPF conditions and euthanized at the indicated time point of an experiment.

**Immunofluorescent analysis of thymic cryosections.** The thymus was fixed overnight in 4% paraformaldehyde (Sigma) at 4 °C, washed three times in PBS, incubated overnight in 30% sucrose at 4 °C, and finally embedded in OCT compound (VWR). Cryoblocks were cut at 8 μm and blocked with PBS containing 5% BSA (w/v) and 0.1% Triton X-100 for 1 hour at room temperature. Samples were incubated overnight at 4 °C with the following primary antibodies: anti-keratin 14, Sirpα, and CD11c-biotin (Fig. 3d) or anti-Involucrin and anti-EpCAM-APC (Supplementary Fig. 3b). The samples were stained with secondary reagents, Goat anti-rat AF-568, goat anti-rabbit AF-647 and streptavidin FITC or goat anti-rabbit AF-488 for one hour at RT. Sections stained only with secondary reagents were used as negative controls. 4′,6-diamino-2-phenylindole (DAPI) was used to visualize cell nuclei. Stained sections were mounted in Vectashield medium (Vector Laboratories) and imaged using a Dragonfly 503 (Andor)—spinning disk confocal microscope with the immersion objective HC PL APO 20×/0.75. A complete list of staining reagents can be found in Supplementary Data 6. Z-stacks were composed using ImageJ and deconvolution was done by Huygens Professional. CD11c$^+$Sirpα$^+$ double positive cells were counted in multiple 300μmx300μm areas in keratin-14 rich (medulla) and keratin-14 negative (cortex) region. Counting was done as a blind experiment by three different investigators. Involucrin$^+$EpCAM$^+$ double positive cells were counted as number of cells per thymic medullary region (determined by DAPI staining).

**Gene expression analysis by qRT-PCR.** Total RNA from FACS-sorted cells was extracted using an RNeasy Plus Micro Kit (Qiagen) and reverse transcribed using RevertAid (ThermoFisher) transcriptase and random hexamers (ThermoFisher). Quantitative RT PCR (qRT PCR) was performed using the LightCycler 480 SYBR Green I Master mix (Roche) on a LightCycler 480 II (Roche). Each sample was tested in duplicate. Threshold cycles were calculated using LightCycler 480 1.5 software. Gene expression was calculated by the relative quantification model[78] using the mRNA levels of the housekeeping gene, Casc3, as a control. Primers were designed using Primer-BLAST (NCBI, NIH). Primers sequences are listed in Supplementary Data 6.

**Bone marrow chimera generation.** Bone marrow cells were isolated from the *femur* and *tibia* of Ly5.1 mice (CD45.1$^+$) and subsequently depleted of erythrocytes using ACK lysis buffer. Recipient mice (Foxn1$^{Cre}$ROSA26$^{TdTOMATO}$, CD45.2$^+$) were irradiated with 6 Gy and reconstituted with $2 \times 10^6$ donor BM cells. These mice were maintained on water supplemented with gentamycin (1 mg/ml) for 10 days. Three weeks after irradiation, the frequency of blood cell reconstitution

was measured by FACS using anti-CD45.1 and CD45.2 antibodies. If the reconstitution was higher than 80%, mice were euthanized 6 weeks after transfer and subjected to further analysis.

**RNA-sequencing and analysis.** mTECs were sorted according to the protocol described above and RNA was extracted using a RNeasy Plus Micro Kit (Qiagen). cDNA synthesis, ligation of sequencing adaptors and indexes, ribosomal cDNA depletion, final PCR amplification and product purification were prepared with a SMARTer® Stranded Total RNA-Seq – Pico Input Mammalian library preparation kit v2 (Takara). Library size distribution was evaluated on a Agilent 2100 Bioanalyzer using the High Sensitivity DNA Kit (Agilent). Libraries were sequenced on a Illumina NextSeq® 500 instrument using a 76 bp single-end high-output configuration resulting in ~30 million reads per sample. Read quality was assessed by FastQC (0.11.9). Subsequent read processing including removing sequencing adaptors (Trim Galore!, version 0.4.5), mapping to the reference genome (GRCm38 (Ensembl assembly version 91)) with HISAT2 (2.1.0), and quantifying expression at the genetic level (featureCounts) was done via the SciLifeLab/NGI-RNAseq pipeline [https://github.com/SciLifeLab/NGI-RNAseq]. Final per gene read counts served as an input for differential expression analysis using a DESeq2 R Bioconductor (3.10). Prior to this analysis, genes that were not expressed in at least two samples were discarded. Genes exhibiting a minimal absolute log2-fold change value of 1 and a statistical significance (adjusted *p*-value < 0.05) between conditions were considered as differentially expressed for subsequent interpretation and visualization. All figures (volcano plots, etc.) were generated using basic R graphical functions. The raw sequencing data were deposited at the ArrayExpress database under accession numbers E-MTAB-8024 (for Fig. 2a, b) and E-MTAB-8025 (for Fig. 2d, e).

**Single-cell RNA sequencing.** DCs were sorted from Foxn1$^{Cre}$ROSA26$^{TdTOMATO}$ as Gr-1$^-$CD11c$^+$TdTOM$^+$ (described in detail in Supplementary Fig. 3a and Fig. 4c). Two independent samples (Sample 1 and 2) were used for further analysis. A single-cell library was prepared by Illumina/Bio-Rad single-cell RNA-seq system with a SureCell WTA 3′ Library Prep Kit according to the manufacturer's instructions. Total cell concentration and viability was ascertained using a TC20 Automated Cell Counter (Bio-Rad). A ddSEQ Single-Cell Isolator (Bio-Rad) was used to co-encapsulate single cells with barcodes and enzyme solutions for cDNA synthesis. Nextera SureCell transposome solution was used for cDNA fragmentation and ligation of sequencing indexes, followed by PCR amplification and short fragment removal. Finally, library fragment length distribution and concentration were analyzed on a Agilent Bioanalyzer 2100 using a High Sensitivity DNA Kit (Agilent). The resulting libraries were sequenced using a 68/75 paired-end configuration on a Illumina NextSeq® 500 instrument resulting in ~73 million reads per sample.

**Single-cell RNA sequencing analysis.** The quality of reads was assessed by FastQC. Cell identification was accomplished with cell barcodes and low-expression cells filtering with UMI-tools[79]. The analysis identified 202 cells in Sample 1 and 218 cells in Sample 2. Reads assigned to the selected cells were mapped to the GRCm38 genome assembly (Ensembl version 91) with HISAT2 (2.1.0). Gene expression was quantified using, featureCounts (2.0.0) after deduplication of per-gene assigned read counts by UMIs with UMI-tools. De-duplicated per-gene read counts were imported into R for exploration and statistical analysis using a Seurat[80] package (version 3.0). Counts were normalized according to total expression, multiplied by a scale factor (10,000), and log-transformed. For cell cluster identification and visualization, gene expression values were also scaled according to highly variable genes after controlling for unwanted variation generated by sample identity. Cell clusters were identified based on t-SNE of the first six principal components of PCA using Seurat's method, FindClusters, with a original Louvain algorithm and resolution parameter value of 0.3. To find cluster marker genes, Seurat's method, FindAllMarkers, along with a likelihood ratio test assuming an underlying negative binomial distribution suitable for UMI datasets was used. Only genes exhibiting a significant (adjusted *p*-value < 0.05) minimal average absolute log2-fold change of 1 between each of the clusters and the rest of the dataset were considered as differentially expressed. For t-SNE expression plots, normalized count data were used. Heatmaps of gene expression per cluster were generated based on gene z-score scaled raw counts. The raw sequencing data was deposited at the ArrayExpress database under accession number E-MTAB-8028.

**In vitro antigen presenting assay.** For the purpose of antigen presentation assay CD14$^+$moDCs were gated as CD11c$^+$MHCII$^+$B220$^-$Xcr1$^-$Cx3cr1$^+$CD14$^+$ and FACS sorted from Aire-HCO mice into DMEM high-glucose medium (Sigma) supplemented with 10% FCS and 1% Penicillin-Streptomycin (Gibco) and cultivated in a 96 well plate together with the A5 hybridoma cell line (HA-specific CD4 T cell hybridoma cells carrying a GFP-NFAT reporter) at a 1:5 ratio (10 000 of CD14$^+$moDC: 50 000 of A5 cells). As a positive control, CD14$^+$moDCs were pulsed with HA peptide (107-119; customized by Thermofisher) at a concentration of 1 μg/ml. After 20 h, the level of GFP expression by A5 hybridomas was analyzed by flow cytometry.

**Induction of T cell transfer colitis and histological analysis**. FACS-sorted $5\times10^5$ TCRβ+CD4+CD45RB^high^CD25− or complete TCRβ+CD4+ were transferred by i. p. injection into Rag1−/− recipient mice (5-7 weeks old). The weight of mice was recorded weekly to monitor the progress of colitis. Mice were euthanized 8 weeks after transfer[51]. Spleens and colons of the animals were weighed and the length of the colon was measured. For histological analysis PBS washed colons were fixed in 4% paraformaldehyde (Sigma) and embedded into paraffin. Tissue sections were cut into 5μm thin slices, deparaffinized, and stained with hematoxylin and eosin (H&E).

**In vitro Tregs suppression assay**. BM-derived DCs (BMDCs) were prepared as follows. BM cells were flushed from femur and tibia of WT C57BL/6J mice and cultured in RPMI media (Sigma) containing 10% FSC and 1% Penicillin/Strepto-mycin (Gibco) supplemented with GM-CSF (5 ng/ml). Fresh media containing GM-CSF was added at day 3 and 5 of cultivation. After 7 days, BMDCs was pulsed with OVA cognate peptide 323-339 (irrelevant OVA 257–264 peptide was used as control) (InvivoGen) at a concentration of 1 μg/ml and co-cultivated with OVA-specific OT-II T cells and Tregs (10 000 BMDCs: 50 000 OT-II T cells: 50 000 Tregs). OT-II T cells were isolated from OT-II+Rag1−/− mice as MACS-enriched CD4+ T cells (CD4+ T Cell Isolation Kit, Miltenyi biotec). CD4+ conventional T cells (Tconv) were used as a negative control. Tregs were isolated from LNs (pLN and mLN) of WT (MyD88^fl/fl^) and MyD88^ΔTECs^ mice using subsequent Auto-MACS (Miltenyi biotec) procedure. CD4-enriched T cells (CD4+ T Cell Isolation Kit, Miltenyi biotec) were stained by anti-CD25 biotin conjugated antibody and CD4+CD25+ Tregs were isolated using Anti-Biotin MicroBeads (Miltenyi biotec). Tconv cells were prepared using Auto-MACS as CD4+CD25− cells. After 3 days of co-cultivation, cells were stained with anti-Vβ5 and anti-Vα2 antibodies to distinguish OT-II+ T cells. Proliferation was measured by FACS using CPD670 staining.

**In vivo model of autoimmune diabetes**. Cd3ε−/−RIP-OVA mice (6–8 weeks old) were intravenously injected by MACS enriched CD8+ T cells ($5\times10^5$ cells per mouse) isolated from lymph nodes and spleen of Rip-OVA Ly5.1 (CD45.1+) mice at day 8. After 7 days (day 1) Cd3ε−/−RIP-OVA mice were intravenously injected, FACS sorted CD4+CD25+ Tregs were isolated from LNs (mLN and pLN) of WT (MyD88^fl/fl^), MyD88^ΔTECs^ mice ($3\times10^5$ cells per mouse), OT-I (OT-I+Rag2−/−, 100 cells per mouse), and OT-II cells (OT-II+Rag1−/−, $1\times10^4$ cells per mouse). BMDCs (generated as described previously, 10 days of culture, media refreshment at day 4 and 7) were pulsed with OVA peptides (OVA 257–264, 2 mM and OVA 323-339, 100 μM, InvivoGen) in the presence of LPS (100 μg/ml, InvivoGen) for 3 h. In all, $1\times10^6$ of antigen-stimulated DCs were used for injection (at day 0). Glucose levels were monitored on a daily basis (between day 5 and 14) using test strips (Diabur-Test 5000, Roche or GLUKOPHAN, Erba Lachema, Czech Republic). The animal was considered to have developed autoimmunity when the concentration of glu-cose in the urine reached ≥10 mmol/l. At day 14, mice were euthanized and the frequency of splenic KLGR1+ OT-1 T cells was measured by flow cytometry.

**Statistical analysis**. The statistical tests used to analyze the data are indicated in figure legends. Graph construction and statistical analysis were performed using Prism 5.04 software (GraphPad). Statistical analysis of RNAseq and scRNAseq data is indicated in the corresponding method section.

**Reporting summary**. Further information on research design is available in the Nature Research Reporting Summary linked to this article.

## Data availability

The authors declare that all data supporting the findings of this study are available within the article and its supplementary information files or from the corresponding author upon reasonable request. The source data underlying Fig. 1c, f, 2c, f, g, 3a–c, e, 4d, f, 5d, f, g, 6a, c, e–i, 7b, d–h and Supplementary Figs. 2b–d, 3a, b, d, e, 4c, 5e, f, 6c, e, h, 7a–d, f, h and 8b, c, e are provided as a Source Data file. The raw RNA sequencing data are deposited at the ArrayExpress database [https://www.ebi.ac.uk/arrayexpress/] under accession numbers E-MTAB-8024 (Fig. 2a, b), E-MTAB-8025 (Fig. 2d, e) and E-MTAB-8028 (Fig. 5a–c).

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

## Acknowledgements

We would like to thank Z. Cimburek and M. Šíma for FACS sorting, Š. Kocourková for preparation of cDNA libraries for RNA sequencing experiments and A. Malinová and I. Novotný for technical assistance with microscopic experiments. V. Kořínek for providing the ROSA26^TdTOMATO and Vav1^Cre mouse models and T. Brdička for OT-II mice. We are indebted to L. Šefc and F. Savvulidi of the Center for Advanced Preclinical Imaging (CAPI) in Prague for their technical assistance with Imaging flow cytometry. We also thank J. Abramson for technical and experimental advice, J. Manning for help with the preparation of the manuscript, and N. Grúňová for graphical design of mice clip arts. This work was supported by Grant 19-23154S from GACR. M.V. was supported by Grant 154215 from GAUK and by Grant ISR-18-31 from the Czech Academy of Sciences. T.B and I.Š. were partially supported by Grant RVO: 68378050-KAV-NPUI. O.S. was supported by SNSF (Promys, IZ11Z0_166538). R.S. was supported by grants LM2015040 and LQ1604 by MEYS) and OP RDI CZ.1.05/1.1.00/02.0109 and CZ.1.05/2.1.00/19.0395 from the MEYS and European Regional Development Fund. L.K. was supported by the European Research Council (ERC-2016-ADG 742290) and the Deutsche Forschungsgemeinschaft (SFB 1054).

## Author contributions

M.V. co-designed and conducted the majority of the experiments and wrote the manuscript. T.B., J.D., and J.B. performed some experiments and provided technical help, I.Š. performed microscopic experiments. A.Č., M.D., and A.A. provided technical support for the work. O.T. and O.Š. performed the experiments using mouse diabetic model. M.K. and V.B. performed RNA sequencing. J.K. analyzed RNAseq and scRNAseq data. R.S. and L.K provided technical and experimental help, mice and material. D.F. designed experiments, supervised research, and edited the paper.

## Competing interests

The authors declare no competing interests.
