## [Peer Review File · Nature Communications]

Reviewers' comments:

Reviewer #1 (DC development, single cell analyses)(Remarks to the Author):

The study by Voboril et al describes the role of TLR expression on Thymic epithelial cells in generating or maintaining the Regulatory T cell compartment, through the recruitment of migratory APCs.

The study is very interesting and raises several questions.

It was shown that antigen delivery occurs also by skin application or subcutaneous injection. In this study, TLR stimulation is performed via intra-thymic injection. Is systemic stimulation sufficient to activate mTECs and induce the recruitment of DCs.

TLR expression is highest on myeloid cells, which upon activation will induce recruitment on effector cells, such as CD14+ monocytes, granulocytes and other cells. In some of the data presented it appears that Foxn1 cre is crossed to Rosa Tomato, However, it is unclear if it is also MyD88 floxed. If not a direct effect on the myeloid compartment can not be excluded.

It would be interesting to look in mTEC myd88 conditional KO mice around birth, when the T regs are formed and when the mouse starts to be exposed to external antigens. Is the T reg compartment already compromised?

The CpG used for the experiment is ODN 1826, which is a type B that strongly activates B cells but not pDCs. Were Type A or Type C also assayed or tried?

Several TLRs appear to be expressed by mTECs, it would be interesting to know if other types of activation leads to similar effects.

MyD88 mediates signaling also for II1, II33, II18 .. Did you check the expression of the other chains? What is known in this regard for mTECs and T reg homeostasis?

Specific Points.

Figure 2

- In figure 2 Some of the most upregulated genes are B cell specific, such as Ig-genes. It is important to have stringent gating strategies and to avoid contaminations, which may influence sequencing results. From the PCA analysis there is strong variation across samples, which type of analysis was performed to implement sample variation?

How was the post-sort flow control before sequencing?

Figure 3

- DC subsets have been recently relabeled as cDC1, cDC2, pDCs and moDCs, please adopt the new nomenclature instead of using a new one. (ref. Guilliams et al 2014 Nat. reviews).

- The majority of Thymic DCs is cDC1, which are characterized by the expression of Cd8a, cDC103 and XCR1.

- In panels B and C, it appears unclear how the subsets change. In Panel B, (untreated mice) pDCs are reduced in MyD88 cKO, tDCs are increased and mDCs reduced. However in Panel D pDCs appear unaltered (black squares and black triangles) as tDCs and mDCs. What are the differences between experiments? Please include the statistics for untreated mice.

- What are the absolute counts for DCs in the thymus? Is the percent reduction of tDCs observed upon CpG injection the results of monocyte recruitment?

- Medullary DCs were shown to be mostly cDC1 and dependent on XCL1 produced by Aire dependent mTECs(Yu et al JEM 2011). How specific is Sirpa staining for mDCs. Could the use of Xcr1 and MHCII

improve the resolution?

Figure 4

- The staining strategy and gating in figure 4E is unclear. As indicated cells are enriched for CD11c. pDCs (left panel) are B220 positive. Are all the cells B220 positive or was there a problem with the staining. Similarly, in the tDCs panel all the cells most of the cells appear to be XCR1 positive, but this is incompatible with the percentage of the other subsets shown.
- The staining for Tdt tomato is vacular, could you include other Markers to exclude ingestion of apoptotic cells.

Figure 5

Is the expression of DC8a on tDC1? What are the identifying genes for each subsets beside the one indicated in panel B. tDC1 does not express Xcr1. Please provide tables for the subsets signature genes. Are mDC2 monocytes?

pDCs are not reduced following treatment with CpG, (Figure 3c), however, specific gating on Tomato positive pDCs shows a significant reduction of pDCs. The other DC subsets behave as shown in figure 2. What mechanisms do you hypothesize for pDCs since single cell analysis does not suggest heterogeneity?

Figure 6

- In Panel F the scale is unclear. Could you please indicate the disease progression as % weight loss? Are the mice losing weight or do WT mice gain weight. What was the age of the mice used?
- Are the T regs generated in the MyD88 KO mice just reduced in numbers or are they functionally different?
- Following injection with CpG an expansion of T regs is observed in WT mice. Are those cells equally capable of T cell inhibition as naturally developed T regs.

Reviewer #2 (Thymic selection, mTEC)(Remarks to the Author):

In this manuscript Voboril et al demonstrate that medullary thymic epithelial cells (mTECs) express several types of Toll-like receptors (TLRs) on a protein level that is comparable to that on dendritic cells (e.g. TLR4 or TLR9). Importantly, rather than being expressed as tissue restricted antigens to which central tolerance is induced, the TLR expression on mTECs seem to play an important functional role in shaping mTEC transcriptional program. Specifically, the authors demonstrate that stimulation of TLR9 on mTECs induces production of various chemokines (Cxcl1, Cxcl2, Cxcl5, Ccl5, etc.). This capacity is diminished in mice expressing mTECs with impaired TLR/Myd88 signaling pathway. The authors further hypothesize that the mTEC-derived and TLR-induced chemokines could potentially attract other immune cells into mTEC proximity and thereby play an important effector role. Indeed, by performing additional experiments the authors demonstrate that these mTEC-derived chemokines are important for recruitment of thymic dendritic cells, including a new subset of CD14+Mgl2-Sirpa+ migratory DCs (mDCs), which were largely diminished in mice with MyD88-deficient TECs. The authors further argue that such recruitment is critical for subsequent transfer of self-antigens from mTECs to DCs for cross-presentation and for subsequent induction of CD25+Foxp3+ Tregs.

In general the manuscript contains very interesting and novel data (the TLR/MyD88 signaling on mTECs, and its role for recruitment of mDCs and the generation of Tregs is highly novel and interesting).

The study has the potential to be published in NC and if some of the data were less correlative and/or the authors would identify the physiological context of TLR9 activation on mTEC, it could have a

potential to be published in even a higher impact journal than NC.

Major points:

1) The authors show that while TEC-specific inactivation results in a significant decrease of thymic Tregs, the ODN injection results in their increase, suggesting that TLR/Myd88 signaling in mTECs shapes mTEC-mediated Treg generation

It should however be noted that there are at least two major subsets of Tregs: – a) de novo generated tTregs (Rag-GFP+, CD73-) and b) recirculating tTregs (RagGFP-, CD73+).

It is therefore necessary to validate whether the impact of TLR signaling is indeed critical for de novo Treg generation in the thymus and/or possibly mediates recirculation of Tregs from the periphery into the thymus. Given that mTECs secrete many different chemokines upon TLR stimulation, the second scenario seems even more likely.

Could the authors dissect what type of tTregs is influenced in the Myd88cKO mice and in the ODN stimulated mice?

2) Most of the figures throughout the manuscript show only frequencies, it would be very informative to show the cell numbers in all such experiments and in particular in Figure 6

3) Although the functional significance of TLR signaling in mTECs is rather convincing and very interesting, the identity of the physiological ligand (context) that would provide such stimulatory signals is missing. Given the high protein expression of TLR9 and TLR4 on mTECs, could the author provide more experimental data on what happens to mTECs during infection by pathogens that induce these receptors. What is the effect of systemic (rather than intrathymic) injection of ODN/LPS on chemokine production by mTECs and tTreg generation/recirculation? Do mice from germ free have similar mTEC/Treg/chemokine phenotype as Myd88 cKO mice.

These experiments could help better understanding whether the ligand for the mTEC-specific TLRs comes from external pathogens or whether it represents an unknown physiological ligand that binds to these receptors in the thymus

4) Colitis experiment

– Could the authors provide more detailed analysis of the intestinal pathology, including results of the histological data.

- It is somewhat unexpected that the Tregs from Myd88 cKO fully mimic the positive control from the colitis model. This would suggest that these Tregs are essentially non-functional. This should be further tested and validated by ex-vivo Treg suppression assay

- In addition to the colitis model, the authors should try to utilize another model of Treg function (self tolerance) in vivo. One possibility would be to treat the Myd88 cKO mice with anti-PD1 mAb, which was recently shown to dramatically worsen the Aire-dependent induction of self tolerance.

5) Discussion – the discussion is a bit long and all over the place; while it is very brief in some of the key issues, such as the identity of the ligand for TLRs. The authors should discuss this in more detail, Rather than repeating the results, the authors should try to discuss the significance of their data as well as

Reviewer #3 (Immune cell development, DC function)(Remarks to the Author):

In this study the authors showed that signaling through TLRs/MyD88 in mTECs is important for the induction of a set of chemokines that recruit migratory mDCs to thymic medulla and facilitate antigen transfer from mTEC to mDCs, which regulate the numbers of CD25+Foxp3+ Tregs cells. Using single-cell RNA-seq analysis the authors identified a new CD14+Mgl2–Sirpa+ mDCs subset and showed that the frequency of this subset was modulated by TLRs/MyD88 signaling in mTECs. The authors also demonstrated that it was this mDC subset that could effectively acquire TEC-derived antigens, decreased cellularity of this CD14+Sirpa+ mDC subset was associated with reduced frequency of thymic CD25+Foxp3+ Tregs. This study therefore described a novel role of mTEC intrinsic TLR/MyD88 signaling in thymic recruitment of mDCs and the generation of Tregs.

This is an interesting study, however several issues need to be clarified:

1. The authors showed in Fig.1 that the expression of TLRs and their signaling adaptors by mTEC and the frequency of mTEC were not affected by the absence of Aire and MyD88, whereas the induction of a group intrathymic chemokines and cytokines was dependent on TLR9/MyD88 signaling in mTEC, which was required for the recruitment of mDCs. As pDC also expressed similar chemokine receptors, one would expect similar level of recruitment should be observed, but the changes in frequency of pDCs in Fig. 3B and Fig3C (without ODN) appeared differently, with Fig.3B displayed a significant decrease in % of pDCs in mice with MyD88 deficient TEC, while the frequency of pDCs in Fig.3C (without ODN) seemed comparable between MyD88fl/fl and MyD88 deficient mice. Similar can be seen for % of mDCs. Can authors provide explanations for this discrepancy? It would be more informative if the authors also show the cell numbers of each DC subsets.
2. Using the single cell RNAseq analysis, the authors identified a new mDC2 subset that can efficiently take up TdTom from mTEC of Foxn1-cre ROSA26-TdTom mice. This mDC2 subset highly expressed Sirpa, Cx3cr1, Cd14, a phenotype very similar to macrophages. Have the authors looked at the expression of CD64 by these cells and excluded the possibility that this subset represented a macrophage rather than mDC population? Can this mDC subset present antigens and activate Ag-specific T cells?
3. The mDCs that migrate into thymus expressed several chemokine receptors, particularly the mDC2 subset highly expressed a different set of chemokine receptors that might be responsible for the increased recruitment of this DC subset upon TLR9/MyD88 signaling in mTEC, have the authors tried to determine which might be the major receptor for mDC2 migration by blocking the interaction of these receptors with their ligands?
4. The Foxn1-cre ROSA26-TdTom mice were used as a model for testing antigen transfer from mTEC to mDCs, however this model can mainly test the transfer of intracellular rather than surface proteins to mDCs. It has been suggested that apoptosis or autophagy of mTEC was involved in transfer of cytoplasmic antigens to DCs, could the TLR9/MyD88 signaling in mTEC enhance these processes? Are MHC molecules involved in this antigen transfer model?
5. Aire+ mTECs are the major source of TSA and self-antigen transfers from these Aire+ mTECs to mDCs are crucial for negative selection of self-reactive thymocytes. It is not clear whether Aire-mediated TSA expression and transfer can also be modulated by TLR9/MyD88 signaling in mTECs?

Point-by-point response to Reviewer's comments:

First, we would like to thank the Reviewers for their constructive comments and questions which has lead to new experiments and analyses which has greatly improved our manuscript.

Reviewer #1 (DC development, single cell analyses) (Remarks to the Author):

The study by Voboril et al describes the role of TLR expression on Thymic epithelial cells in generating or maintaining the Regulatory T cell compartment, through the recruitment of migratory APCs. The study is very interesting and raises several questions.

- We thank the Reviewers for this positive comment.

It was shown that antigen delivery occurs also by skin application or subcutaneous injection. In this study, TLR stimulation is performed via intra-thymic injection. Is systemic stimulation sufficient to activate mTECs and induce the recruitment of DCs

- We have now included a novel set of data showing that repeated intraperitoneal (i.p.) injections of CpG ODN (TLR9 ligand) is not sufficient for the upregulation of chemokines associated with TLR/MyD88 signaling in mTECs^{high} observed upon intrathymic administration of this ligand (Supplementary Fig. 2c). As suggested by the reviewer, we also tested whether the systemic stimulation of TLR9 would induce the recruitment of DCs into the thymus. As shown below, and in a full agreement with the unchanged levels of mTEC-derived chemokines, the frequencies of thymic DC-populations were not affected. This suggests that the ligand which triggers TLR responses in mTECs is likely of intrathymic in nature, and not induced by or associated with a systemic TLR9 stimulation.

TLR expression is highest on myeloid cells, which upon activation will induce recruitment on effector cells, such as CD14⁺ monocytes, granulocytes and other cells. In some of the data presented it appears that Foxn1 cre is crossed to Rosa Tomato, However, it is unclear if it is also MyD88 floxed. If not a direct effect on the myeloid compartment cannot be excluded.

- This is indeed an essential point. This issue has been clarified in Figure 3c, where we used MyD88^{ΔTECs} animals, in which TLR signaling has been abrogated specifically in TEC cells and not in other thymic cellular compartments, including the myeloid compartment. Thus, neither myeloid nor other thymic cellular compartments are able to compensate for the lack of MyD88 signaling in TECs. This in particular concerns the expression of a set of chemokines which leads to increased recruitment of CD14⁺ moDCs. In agreement, and as far as CAT is concerned, we showed that in Foxn1^{Cre}ROSA26^{TdTom} animals the frequencies of TdTom⁺ cDC1 (decreased) and cDC2 (increased) upon CpG ODN stimulation were altered (Fig. 5d). Importantly, such alterations are fully recapitulated in the thymus of TLR9 stimulated WT animals (Fig. 3c). This strongly suggests that CAT directly correlates with the frequencies of given subtypes in the thymus, which in turn are regulated by MyD88 signaling in TECs. We concur that a triple transgenic mouse model with Foxn1^{Cre}MyD88^{fl/fl}TdTom genotype stimulated with CpG ODN would be the ultimate experiment to confirm this conclusion. However, despite our huge effort, we do not have these mice in hand to perform this type of experiment.

It would be interesting to look in mTEC myd88 conditional KO mice around birth, when the T regs are formed and when the mouse starts to be exposed to external antigens. Is the T reg compartment already compromised?

- We thank the reviewer for this question. We have now included a novel set of data where we compared the frequencies and numbers of thymic CD25⁺Foxp3⁺ Tregs in MyD88^{ΔTECs} and WT (MyD88^{fl/fl}) newborns (4 days old) mice as well as in the Germ-free (GF) and Specific-pathogen-free (SPF) adult (3 weeks old) WT mice. We found no significant changes in Treg numbers, in newborn mice from MyD88^{ΔTECs}, or GF mice compared to their WT counterparts (Supplementary Fig. 7b, c). This suggests that the MyD88 signaling involved in the generation of Tregs is likely mediated by the ligand of endogenous, as opposed to exogenous (bacterial) origin. Also, as suggested by the Reviewer #2, we tested whether the TLR/MyD88 signaling in mTECs affected the production of newly generated Tregs (CD73⁻) or the recirculation of peripheral Tregs (CD73⁺) back to the thymus. We found that only the former was affected in MyD88^{ΔTECs} (Fig 6d, e). Together, this indicates that TLR/MyD88 signaling in mTECs does not affect the first wave of Tregs generation but rather the later one (from 3-6 weeks mice age) (Yang et al., Science 2015).

The CpG used for the experiment is ODN 1826, which is a type B that strongly activates B cells but not pDCs. Were Type A or Type C also assayed or tried?

- Since we hypothesized, that it is the TLR9 signaling in mTECs which affects the migration and/or activation of thymic DC populations, we deliberately utilized type B ODN 1826, which as pointed by the reviewer, does not activate pDCs on its own. Thus, intrathymic CpG ODN injection allowed direct observation of the phenotype caused by the activation of mTECs and not by the pDCs. For this reason, the ODN of type A or C was not used in this work.

Several TLRs appear to be expressed by mTECs, it would be interesting to know if other types of activation leads to similar effects.

- We thank the reviewer for raising this point. As we have shown in Fig 1, several TLRs are expressed by mTECs, with TLR9 seemingly being the receptor with the highest expression. To test the assumption that also other TLRs are functionally wired to intracellular signaling, we *in vitro* stimulated mTECs^{high} cells with LPS (the ligand of TLR4). We found that TLR4 stimulation resulted in the upregulation of several chemokines which are associated with TLR9 stimulation, albeit at a lower level (Supplementary Fig 2d). This suggests that the activation of TLRs expressed in mTECs^{high} leads to similar changes in the genetic expression profile, particularly of chemokines, although a more robust experimental approach and analysis to illustrate this general concept would be necessary.

MyD88 mediates signaling also for Il1, Il33, Il18 .. Did you check the expression of the other chains? What is known in this regard for mTECs and T reg homeostasis?

- As suggested by the reviewer, we compared the expression levels of receptors for cytokines of the IL-1 family (*Il1r1*, *Il1rl1* and *Il18r1*) and found that these genes are expressed by mTECs^{high} at comparable levels as TLR9 (Supplementary Fig. 3a). Since very little is known about the role of these chemokines in the mechanisms of central tolerance, we tested whether the *in vitro* stimulation of mTECs by these cytokines could trigger the same response as TLR9 stimulation. As shown in Supplementary Fig. 3b, only the *in vitro* stimulation with IL-1 β lead to a comparable affect. This data suggests that IL-1 β could act as a co-regulator of chemokines and cytokine expression in mTECs^{high}, even though proving or disproving this conclusion would still require careful and thorough investigation, which goes beyond the scope of this manuscript. At any rate, these interesting results concerning IL-1 β stimulation are discussed in the context of TLR9/MyD88 signaling in more detail in the Discussion chapter.

Specific Points:

Figure 2

- In figure 2 some of the most upregulated genes are B cell specific, such as Ig-genes. It is important to have stringent gating strategies and to avoid contaminations, which may influence sequencing results. From the PCA analysis there is strong variation across samples, which type of analysis was performed to implement sample variation? How was the post-sort flow control before sequencing?

- We thank the review for raising this point. We carefully tested whether mTECs^{high}, sorted as shown in Supplementary Fig. 1a, could be potentially contaminated by B-cells. In the figure

projecting gates used for sorting, the possible B-cell contamination in

EpCAM⁺CD11c⁻Ly51⁻MHCII⁺CD80⁺ mTEC gate is negligible. Since B-cell specific Ig-genes have also been found in a single-cell RNAseq analysis of TECs (Bornstein et al., Nature 2018) and due to the fact, that mTECs^{high} express more than 80% of the murine protein coding genome (Sansom et al., Genome Research 2014), we suggest

that our data reflects the fact that mTECs express those genes intrinsically, as a consequence of ectopic gene expression, a typical feature of these cells.

In the case of PCA analysis of mTECs from MyD88^{ΔTECs} and WT (MyD88^{fl/fl}) mice (Fig. 2a), we respectfully disagree with the reviewer's opinion that there is a strong variation across samples. As seen from the PCA analysis, the major differences are clearly visible on PCA1 axis (56% variance) which represents the phenotype of MyD88 depletion. On the other hand, the variations between samples are visible mostly on PCA2 axis, which shows only very limited variability (16% variance) compared to PCA1. This demonstrates that intra-sample variations are small compared

to the variability that is caused by the depletion of MyD88.

As requested by the reviewer, we have shown here the post-sort re-analysis that was done for each sort of mTECs. Additionally, only clean samples (as shown here) were used for further analysis.

Figure 3

- DC subsets have been recently relabeled as cDC1, cDC2, pDCs and moDCs, please adopt the new nomenclature instead of using a new one. (ref. Guilliams et al 2014 Nat. reviews).

- As suggested by the reviewer, we implemented a new nomenclature of the mononuclear phagocyte system based on Guilliams et al., Nature Reviews Immunology 2014. Specifically, tDCs are now referred to as cDC1 and mDCs as cDC2. This nomenclature has been clearly explained in the manuscript.

- *The majority of Thymic DCs is cDC1, which are characterized by the expression of Cd8a, cDC103 and XCR1.*

- This question relates to the way the cells have been gated. We agree that among thymic cDCs, gated as CD11c^{high}MHCII^{high}, the majority of cells are Xcr1⁺Sirpα⁻ cDC1. However, as shown by our scRNA seq, the populations of thymic DCs are quite heterogeneous even by the expression of CD11c and MHCII. For this reason, we implemented a much more encompassing gating on CD11c and MHCII, which does not exclude monocytes, moDCs, and DC-precursors. Since many of these subsets also express Sirpα, the Sirpα⁺ population in our hands represents the majority of thymic DC population if gated as CD11c^{int/+}MHCII^{int/+}. This phenomenon was also observed and implemented by others (Lancaster et al. Nature Communications 2019).

- *In panels B and C, it appears unclear how the subsets change. In Panel B, (untreated mice) pDCs are reduced in MyD88 cKO, tDCs are increased and mDCs reduced. However in Panel D pDCs appear unaltered (black squares and black triangles) as tDCs and mDCs. What are the differences between experiments? Please include the statistics for untreated mice.*

- We thank for pointing out this discrepancy. To resolve this issue, we re-analyzed the data and measured more samples (+ 2 mice for each condition). The updated graphs now clearly show the same trends as in Fig. 3b. However, it is important to realize that the experimental settings in these two panels are different. In contrast to Fig. 3b, all mice from Fig. 3c were anesthetized and intrathymically injected with PBS or CpG ODN. Since the injection of PBS alone could potentially cause local tissue damage and affect the frequencies of certain DCs populations in the thymus, the samples show much higher variability in the frequencies DCs populations, than in the case of non-manipulated thymi used for the Fig. 3b. The p-values have been now included into the revised Fig. 3c. We have also added the statistical information regarding untreated mice as requested.

- *What are the absolute counts for DCs in the thymus? Is the percent reduction of tDCs observed upon CpG injection the results of monocyte recruitment?*

- Unfortunately, at this stage, we are unable to provide the total numbers of DC and DC subsets per thymus. We can only quantify the number of MHCII⁺CD11c⁺ DCs, as well as pDCs, cDC1, and cDC2 as provided in a new Supplementary Fig. 4c. The reason for this is as follows: to know the absolute numbers of DCs per entire thymus, one must analyze the entire thymic cell population or its reasonably bulky aliquot. In addition, staining cells with a panel of antibodies for 9-parametric FACS analysis to get info about all distinct thymic DC subsets (representing in total only 0,4% of all thymocytes) is quite ineffective and provides fuzzy data. To overcome this limitation and obtain a reproducible staining pattern, we had to enrich for all CD11c⁺ cells and only then perform the staining. Because this approach usually yields a substantial proportion of all thymic DCs (≤60% compared to their reported absolute numbers, Hu et al, Cell Reports, 2017), it still enables us to perform a broad comparison of its subset cellularities within the DC compartment. Since FACS analysis was performed on the majority of such CD11c⁺ enriched DCs, our data strongly suggests that the increase of CD14⁺moDCs is indeed accompanied with a decreased cellularity of tDCs. However, even if we agree with the assumption that the absolute number of tDCs remained unchanged, then the number of CD14⁺moDCs has been definitively increased. Essentially, this is the most important conclusion of these experiments, since the change

of the ratio between the frequencies of cDC2 to cDC1 is a defining factor in the regulation of Treg production (see the Discussion section). Thus, we believe that enumerating DC subsets in this manner is fully sufficient to substantiate the effect of mTEC-mediated TLR9-dependent signaling on the enhanced recruitment of CD14⁺moDCs.

- Medullary DCs were shown to be mostly cDC1 and dependent on XCL1 produced by Aire dependent mTECs (Yu et al JEM 2011). How specific is Sirpa staining for mDCs. Could the use of Xcr1 and MHCII improve the resolution?

• As alluded to above and shown in Fig. 5a, the Sirpa⁺ DC-gate contains at least two different subpopulations of DCs: Mgl2⁺cDC2 (previously called mDC1 in our original manuscript) and CD14⁺moDC (previously mDC2), which are phenotypically, developmentally, and functionally different (Fig. 5b). Hence, Mgl2 and CD14 should be used as additional markers to Sirpa⁺ to distinguish conventional cDC2 from moDCs. As suggested by the reviewer, we tested whether Xcr1 and MHCII staining would improve the resolution. The back-gating of Xcr1⁺ (green gate) or Sirpa⁺ (blue gate) DC subsets into Xcr1 vs MHCII gating, shows that the Xcr1⁻MHCII⁺ population also contains the Sirpa⁻ cells (red gate). This clearly demonstrates that Sirpa, Mgl2, and CD14 staining can be used for the analysis of Xcr1⁻ thymic DC populations.

Figure 4

- The staining strategy and gating in figure 4E is unclear. As indicated cells are enriched for CD11c. pDCs (left panel) are B220 positive. Are all the cells B220 positive or was there a problem with the staining. Similarly, in the tDCs panel all the cells most of the cells appear to be XCR1 positive, but this is incompatible with the percentage of the other subsets shown.

• We believe that this is a misunderstanding because the flow cytometry plots shown in Fig. 4 depict the frequency of TdTOM⁺ cells among the pre-gated population of DCs. To clarify the data generated in this figure, we replaced the plots with histograms showing the exact frequency of TdTOM⁺ cells among thymic pDC, cDC1, and cDC2 populations (Fig. 4e).

- The staining for Tdt tomato is vacular, could you include other Markers to exclude ingestion of apoptotic cells.

- We thank reviewer for this suggestion. The main purpose of this figure was to visualize the transfer and association of the TdTOM antigen with CD11c⁺ cells using the ImageStream approach. It was recently described that the antigen transfer from mTECs to thymic DCs occurs mostly by the scavenging of apoptotic bodies by myeloid cells (Perry et al, Immunity 2018). In this context, the transfer of TdTOM from Foxn1^{Cre}Rosa26^{TdTOMATO} TECs on CD11c⁺DCs in our experiments is visually consistent with such a conclusion. As suggested by the reviewer, we agree that using other markers, such as LysoTracker, could help to better resolve this issue. Although this would be very interesting, we feel that such experimentation would digress from the main topic of this manuscript being out of the scope of this current study. Indeed, more focused studies on this subject will be required to address this potentially complex issue.

Figure 5

-Is the expression of DC8a on tDC1? What are the identifying genes for each subsets beside the one indicated in panel B. tDC1 does not express Xcr1. Please provide tables for the subsets signature genes. Are mDC2 monocytes?

- Thank you very much for these important questions and suggestions. We believe, that resolving these issues has helped improve the quality of the manuscript. We have included a new heat map analysis of genes that determine the nature of each cluster defined by our scRNA seq of TdTOM⁺CD11c⁺ thymic cells (see Fig. 5b). We also added a new Supplementary Table 5 showing all differentially expressed genes among the DC clusters. As shown in Fig. 5b, Cd8a is expressed mostly by pDCs but also by cDC1b (previously called tDC2) and at lower levels by cDC1a (previously called tDC1). cDC1a were defined as cDC1 cells according to the expression of *Batf3* (Hildner et al., Science 2008) and *Ly75*. These cDC1a also share many markers with the previously defined thymic *Xcr1*^{low}*Ccr7*⁺ DCs (Ardoulin et al., Immunity 2016) such as *Ccr7*, *Relb*, *Ccl5*, *Il15r*, *Cd40*). Based on this, we defined this population as cDC1a.

In the case of the previously described population of mDC2, we have included novel data which helps identify these cells as monocyte-derived DCs (moDC). As shown in Fig. 5b, these cells share many markers with cDCs (*Itgax*, *Irf4*, *Sirpa* and *Itgam*), “classical” tissue resident macrophages (*Mertk*, *Mafb* and *Lyz2*), and with monocytes (*Ly6c1*, *Ly6c2*, *Fn1*, *Ccr2* and *Ifitm3*). Since these cells are able to process and present antigen and activate antigen-specific T-cells (Supplementary Fig. 6b, c), we have referred to our previously defined population of mDC2 as CD14⁺ moDC.

-pDCs are not reduced following treatment with CpG, (Figure 3c), however, specific gating on Tomato positive pDCs shows a significant reduction of pDCs. The other DC subsets behave as shown in figure 2. What mechanisms do you hypothesis for pDCs since single cell analysis does not suggest heterogeneity?

- This point is well taken. As described in Fig. 4e and f, the ability of pDCs, in comparison to cDC1 or cDC2, to acquire the TEC-derived antigen is limited. Also, it is known that TECs constitute a relatively rare cell population of thymic cells (Klein et al. 2009) and the number of TdTOM⁺TECs is not increased after CpG ODN stimulation (Supplementary Fig. 3d). This suggests that the amount of antigen that can be potentially transferred to DCs is fairly limited and is mostly captured by increased numbers of CD14⁺moDCs. This could explain the fact, that even

though the entire population of thymic pDCs is not affected by intrathymic TLR9 stimulation, the frequency of TdTOM⁺ pDCs is significantly decreased because due to competition for the access to antigen transfer by increased CD14⁺ moDCs after CpG ODN stimulation.

Figure 6

- In Panel F the scale is unclear. Could you please indicate the disease progression as % weight loss? Are the mice losing weight or do WT mice gain weight. What was the age of the mice used?

- Disease progression was depicted as the percentage of mouse original weight (Fig. 7b). The host that received CD4⁺ T-cells from WT mice continuously gained weight over time. Those mice that received CD4⁺ T-cells from MyD88^{ΔTECs} or positive controls ceased to gain weight approximately 4 weeks after adoptive transfer and started to lose weight at week 7. The age of mice used for T-cell transfer colitis model has been added to the method section. Specifically, mice from 5 to 7 weeks of age were used.

- Are the T regs generated in the MyD88 KO mice just reduced in numbers or are they functionally different?

- This is indeed an important question. We have now included novel data which shows the reduced expression of CD25 by Tregs from MyD88^{ΔTECs} (Fig. 7f) and their reduced capacity to suppress the T-cells proliferation *in vitro* (Fig. 7g). We have also shown that Tregs that were isolated from MyD88^{ΔTECs} have the reduced capacity to prevent the onset of diabetes in the RIP-OVA mouse model system (Supplementary Fig. 8c, d, e).

- Following injection with CpG an expansion of T regs is observed in WT mice. Are those cells equally capable of T cell inhibition as naturally developed Tregs.

- This is potentially very interesting question. However, since we found, that the CD25⁺Foxp3⁺ population of thymic Tregs which was upregulated in response to TLR9 intrathymic stimulation consists of CD73⁻ newly generated Tregs as well as CD73⁺ thymically recruited recirculating ones (see Fig. 6g), we did not address this question. The uncertainty that accompanies such an experiment is that it would test the suppressive capacity of the population of newly generated, MyD88 signaling-induced Tregs mixed with recirculating Tregs which have been generated under unperturbed conditions. This in our opinion would be incomparable with the situation where all Tregs are generated under unperturbed conditions. We believe that interpretations of such results would be unclear and inconclusive.

Reviewer #2 (Thymic selection, mTEC) (Remarks to the Author):

In this manuscript Voboril et al demonstrate that medullary thymic epithelial cells (mTECs) express several types of Toll-like receptors (TLRs) on a protein level that is comparable to that on dendritic cells (e.g. TLR4 or TLR9). Importantly, rather than being expressed as tissue restricted antigens to which central tolerance is induced, the TLR expression on mTECs seem to play an important functional role in shaping mTEC transcriptional program. Specifically, the authors demonstrate that stimulation of TLR9 on mTECs induces production of various chemokines (Cxcl1, Cxcl2, Cxcl5, Ccl5, etc.). This capacity is diminished in mice expressing mTECs with impaired TLR/Myd88 signaling pathway. The authors further hypothesize that the mTEC-derived and TLR-induced chemokines could potentially attract other immune cells into mTEC proximity and thereby

play an important effector role. Indeed, by performing additional experiments the authors demonstrate that these mTEC-derived chemokines are important for recruitment of thymic dendritic cells, including a new subset of CD14+Mgl2–Sirpa+ migratory DCs (mDCs), which were largely diminished in mice with MyD88-deficient TECs. The authors further argue that such recruitment is critical for subsequent transfer of self-antigens from mTECs to DCs for cross-presentation and for subsequent induction of CD25+Foxp3+ Tregs.

In general the manuscript contains very interesting and novel data (the TLR/MyD88 signaling on mTECs, and its role for recruitment of mDCs and the generation of Tregs is highly novel and interesting). The study has the potential to be published in NC and if some of the data were less correlative and/or the authors would identify the physiological context of TLR9 activation on mTEC, it could have a potential to be published in even a higher impact journal than NC.

- We thank Reviewer #2 for highlighting the novelty and general interest of our work. We agree that some of our conclusions are of correlative nature. We concur that the knowledge of “*physiological context of TLR9 activation on mTECs*” would increase the impact of the study. Towards this, we have performed a number of additional experiments which touch upon this issue. In particular, we have tested the potential source and origin of the ligand triggering the mTEC-dependent response using Germ-free mice as suggested by this reviewer. We believe that this novel data will provide fresh insight into this issue and improve the quality of the manuscript.

Major points:

1) The authors show that while TEC-specific inactivation results in a significant decrease of thymic Tregs, the ODN injection results in their increase, suggesting that TLR/Myd88 signaling in mTECs shapes mTEC-mediated Treg generation. It should however be noted that there are at least two major subsets of Tregs: – a) de novo generated tTregs (Rag-GFP+, CD73-) and b) recirculating tTregs (RagGFP-, CD73+). It is therefore necessary to validate whether the impact of TLR signaling is indeed critical for de novo Treg generation in the thymus and/or possibly mediates recirculation of Tregs from the periphery into the thymus. Given that mTECs secrete many different chemokines upon TLR stimulation, the second scenario seems even more likely. Could the authors dissect what type of tTregs is influenced in the Myd88cKO mice and in the ODN stimulated mice?

- We thank reviewer for this important insight. We have now included novel data comparing the numbers of newly generated, CD73⁻CD25⁺Foxp3⁺ Tregs and recirculating CD73⁺CD25⁺Foxp3⁺ in MyD88^{ΔTECs} as well as in TLR9 intrathymically stimulated mice (see Fig. 6d, e, g). Our results clearly show that the abrogation of MyD88 signaling in mTECs primarily affects the generation of CD25⁺Foxp3⁺ thymic Tregs and not the recirculation of peripheral Tregs back to the thymus. On the other hand, the intrathymic injection of CpG ODN lead to the increase in number of not only CD73⁻ newly generated Tregs but also recirculating CD73⁺. As alluded to by the reviewer, we suggest that the increased recirculation of Tregs after TLR9 stimulation was mediated mostly by the upregulation of Ccl20 in mTECs, the cytokine which has been previously shown to be essential for Tregs recirculation (Cowan et al., Eur J Immunol, 2018).

Since we predict, that the increased generation of CD73⁻ Tregs after TLR9 stimulation is mediated by antigen presentation by DCs, we also intrathymically injected CpG ODN into H2-Ab1^{fl/fl}CD11c^{Cre} (H2-Ab1^{ΔDCs}) mice, where antigen presentation by DCs is abrogated (see Fig. 6 h, i). We found that while the increased generation of Tregs is abrogated in these mice, we observed

that Tregs recirculation was not affected. This implies that MHCII presentation by DCs has a negligible effect on the increased recirculation of Tregs after TLR9 stimulation but is essential for *de novo* generation of Tregs.

2) Most of the figures throughout the manuscript show only frequencies, it would be very informative to show the cell numbers in all such experiments and in particular in Figure 6.

- We thank for this point. Where possible, the cell numbers are now indicated in the manuscript. The cell numbers associated with Fig. 6 have been added to Supplementary Fig. 7 (related to Fig. 6).

3) Although the functional significance of TLR signaling in mTECs is rather convincing and very interesting, the identity of the physiological ligand (context) that would provide such stimulatory signals is missing. Given the high protein expression of TLR9 and TLR4 on mTECs, could the author provide more experimental data on what happens to mTECs during infection by pathogens that induce these receptors. What is the effect of systemic (rather than intrathymic) injection of ODN/LPS on chemokine production by mTECs and tTreg generation/recirculation? Do mice from germ free have similar mTEC/Treg/chemokine phenotype as Myd88 cKO mice. These experiments could help better understanding whether the ligand for the mTEC-specific TLRs comes from external pathogens or whether it represents an unknown physiological ligand that binds to these receptors in the thymus

- An excellent point, we appreciate this inquiry. To gain some insight, and attempt to resolve this issue (at least partly), we tested whether systemic injection of CpG ODN affected the production of chemokines by mTECs which, in turn, subsequently increases the migration of CD14⁺moDC into the thymus correlating with boosting the generation/recirculation of Tregs. As described in the answer to the question 1, Reviewer #1, we performed repeated i.p. injection of CpG ODN and found that this type of stimulation was not sufficient for both the upregulation of relevant chemokines by mTECs (Supplementary Fig. 2c) and increased migration of DCs into the thymus. For these reasons, we would expect that the number of thymic Tregs would be unchanged.

We also tested whether the set of chemokines/cytokines which have been downregulated in mTECs from MyD88^{ΔTECs} mice would also be decreased in mice from Germ-free (GF) relative to Specific-pathogen-free (SPF) conditions. As shown in Supplementary Fig. 2b, the lack of bacterial-derived MyD88 signals (GF mice) had no effect on the expression level of these chemokines and cytokines in WT mice. This was further corroborated by the fact that GF mice displayed normal numbers of Tregs (also in Owen et al., Nature Immunology 2019) (Supplementary Fig. 7c). Since this data strongly suggests that the ligand which triggers TLR9/MyD88 signaling in mTECs^{high} is of endogenous thymic-derived origin, further experiments would be needed to better understand its physiology and nature. This would also be applicable to other potential TLR ligands.

4) Colitis experiment– Could the authors provide more detailed analysis of the intestinal pathology, including results of the histological data.

- Additional histological analysis of colon tissue have been added, as well as the comparison of colon weight/length ratio is now included in main figure (Fig. 7c, d).

- It is somewhat unexpected that the Tregs from Myd88 cKO fully mimic the positive control from the colitis model. This would suggest that these Tregs are essentially non-functional. This should be further tested and validated by ex-vivo Treg suppression assay

- We thank reviewer for this suggestion. As described in our reply this question posed by Reviewer #1, the Tregs from MyD88^{ΔTECs} have a significantly reduced capacity to suppress the proliferation of antigen specific T-cells *in vitro* (Fig. 7g). This suggests that not only the reduced numbers of Tregs but also their impacted functionality additively and/or synergistically contribute to the development of the colitis phenotype caused by the transfer of CD4 T-cell from MyD88^{ΔTECs} mice.

- In addition to the colitis model, the authors should try to utilize another model of Treg function (self tolerance) in vivo. One possibility would be to treat the Myd88 cKO mice with anti-PD1 mAb, which was recently shown to dramatically worsen the Aire-dependent induction of self tolerance.

- We appreciate this challenging suggestion, however, as our specific focus was on the functionality of Tregs from MyD88^{ΔTECs} mice, we would prefer to explore an alternative *in vivo* mouse model of diabetes. We have taken advantage of a system in which the injection of OVA specific OT-I and OT-II T-cell into T-cell depleted RIP-OVA mice lead to the development of diabetes (see the Method section). Also, it is known that co-injection of WT Tregs into the host could postpone the induction of diabetes. As shown in Supplementary Fig. 8c, Tregs from MyD88^{ΔTECs} mice show a reduced capacity to prevent the early onset of diabetes compared to Tregs from WT mice. Consistent with this observation, the OT-I T-cells that were isolated from the diabetic mouse model that received the MyD88^{ΔTECs} Tregs showed a much higher expression of KLRG1, the marker of effector CD8⁺ T-cells.

5) Discussion – the discussion is a bit long and all over the place; while it is very brief in some of the key issues, such as the identity of the ligand for TLRs. The authors should discuss this in more detail, Rather than repeating the results, the authors should try to discuss the significance of their data as well as

We thank reviewer for this point. The discussion has been rewritten, enriched with a set of data related to new experiments added to the manuscript during the revision process.

- **Reviewer #3 (Immune cell development, DC function) (Remarks to the Author):**

In this study the authors showed that signaling through TLRs/MyD88 in mTECs is important for the induction of a set of chemokines that recruit migratory mDCs to thymic medulla and facilitate antigen transfer from mTEC to mDCs, which regulate the numbers of CD25+Foxp3+ Tregs cells. Using single-cell RNA-seq analysis the authors identified a new CD14+Mgl2–Sirpa+ mDCs subset and showed that the frequency of this subset was modulated by TLRs/MyD88 signaling in mTECs. The authors also demonstrated that it was this mDC subset that could effectively acquire TEC-derived antigens, decreased cellularity of this CD14+Sirpa+ mDC subset was associated with reduced frequency of thymic CD25+Foxp3+ Tregs. This study therefore described a novel role of mTEC intrinsic TLR/MyD88 signaling in thymic recruitment of mDCs and the generation of Tregs. This is an interesting study, however several issues need to be clarified:

- We thank Reviewer #3 for pointing out that our study is novel and interesting. As with Reviewer #1, Reviewer #3 points out that the “CD14⁺mDC2” share several markers with monocytes and macrophages. We feel that suggestions from both reviewers has helped guide us to a more

improved manuscript. Specifically, to clarify the phenotype, origin, and function of these cells, we performed several additional experiments and analyses which lead us to redefine them as monocyte-derived DCs (CD14⁺moDC).

1. The authors showed in Fig.1 that the expression of TLRs and their signaling adaptors by mTEC and the frequency of mTEC were not affected by the absence of Aire and MyD88, whereas the induction of a group intrathymic chemokines and cytokines was dependent on TLR9/MyD88 signaling in mTEC, which was required for the recruitment of mDCs. As pDC also expressed similar chemokine receptors, one would expect similar level of recruitment should be observed, but the changes in frequency of pDCs in Fig. 3B and Fig3C (without ODN) appeared differently, with Fig.3B displayed a significant decrease in % of pDCs in mice with MyD88 deficient TEC, while the frequency of pDCs in Fig.3C (without ODN) seemed comparable between MyD88fl/fl and MyD88 deficient mice. Similar can be seen for % of mDCs. Can authors provide explanations for this discrepancy? It would be more informative if the authors also show the cell numbers of each DC subsets.

- As described in detail to the question asked by Reviewer #1 (related to Fig. 3), we performed additional experiments and updated the graphs in Fig. 3c. This figure now clearly shows the very same trends that are seen in Fig. 3b. However, and as pointed out above, the intrathymic injection of PBS could potentially affect the frequency of certain DCs populations in the thymus, thus increasing the variability among samples.

2. Using the single cell RNAseq analysis, the authors identified a new mDC2 subset that can efficiently take up TdTOM from mTEC of Foxn1-cre ROSA26-TdTOM mice. This mDC2 subset highly expressed Sirpa, Cx3cr1, Cd14, a phenotype very similar to macrophages. Have the authors looked at the expression of CD64 by these cells and excluded the possibility that this subset represented a macrophage rather than mDC population? Can this mDC subset present antigens and activate Ag-specific T cells?

- We thank for reviewer for this suggestion. Towards this end, we have now included a novel heat map analysis of signature genes that determines the nature of each cluster defined by our scRNA seq of TdTOM⁺CD11c⁺ thymic cells (see Fig. 5b). This analysis has helped us to describe several interesting features of CD14⁺moDCs (previously “CD14⁺mDC2 population”): (i) these cells do not express *Flt3*, the receptor for Flt3-ligand, which is crucial for the differentiation of classical DCs from the monocyte/macrophage-DC progenitor (MDP) (Waskow et al., Nature Immunology 2008), whereas the receptors for Csf2, i.e. *Csfr2a* and *Csfr2b*, are expressed at very high levels; (ii) The expression profile of these cells is enriched for genes that are associated with monocyte/macrophages, such as *Cx3cr1*, *Mertk*, *Cebpb*, *Fcgr2b*, *Fcgr3* or *Lyz2* specifically circulating monocytes as *Ly6c1*, *Ly6c2*, *Cd14*, *Ccr2* or *Fn1*; and (iii) As shown in Supplementary Fig. 6b, these cells also express genes which are related to processing and presentation of antigen (*Ctsc*, *Cd74*, *H2-Ab*, *H2-Aa*, *Cd86*). As asked by this reviewer, we also tested the expression of CD64, a marker which denotes monocytes-derived and classical DCs (Langlet et al., Journal of Immunology, 2012). We found that *Fcgr1* (Cd64) was indeed predominantly expressed by CD14⁺moDCs. This information strongly suggests that this population is of monocyte-derived origin rather than representing the classical DC population.

As suggested by the reviewer, we tested whether these cells were able to present antigens and activate antigen specific T-cells. We have now included novel data showing the capacity of the

CD14⁺moDCs cell to present mTEC-produced antigens and activate antigen-specific T-cells (see Supplementary Fig. 6c).

3. The mDCs that migrate into thymus expressed several chemokine receptors, particularly the mDC2 subset highly expressed a different set of chemokine receptors that might be responsible for the increased recruitment of this DC subset upon TLR9/MyD88 signaling in mTEC, have the authors tried to determine which might be the major receptor for mDC2 migration by blocking the interaction of these receptors with their ligands?

- Indeed, this is a very interesting question. To determine the receptor responsible for CD14⁺moDCs migration to the thymus, we crossed Cxcr2^{fl/fl} mice with the pan-hematopoietic driver, Vav1^{Cre}, to abrogate the signaling of its cognate ligands Cxcl1, 2, 3, and 5 chemokines that were among the most upregulated genes in mTECs after TLR9 stimulation. However, we observed no changes in CD14⁺moDC migration after TLR9 stimulation between Cxcr2^{fl/fl}Vav1^{Cre} mice or WT controls (see Supplementary Fig. 6h). This points to the fact that the ligands of Ccr3 or Ccr5 (Ccl3, Ccl4, Ccl5 or Ccl24) potentially regulate the entry of moDCs into the thymic medulla. On the other hand, since most of these receptors can be triggered by many different ligands (Griffith et al., Annu Rev Immunol 2014) and the receptors for these chemokines show a high level of redundancy (Dyer et al., Immunity 2019), the determination of a single chemokine/chemokine receptor would be at this point extremely difficult.

4. The Foxn1-cre ROSA26-TdTom mice were used as a model for testing antigen transfer from mTEC to mDCs, however this model can mainly test the transfer of intracellular rather than surface proteins to mDCs. It has been suggested that apoptosis or autophagy of mTEC was involved in transfer of cytoplasmic antigens to DCs, could the TLR9/MyD88 signaling in mTEC enhance these processes? Are MHC molecules involved in this antigen transfer model?

- This is one of the emerging and also most intriguing questions. In general, due to the heterogeneity of antigen-presenting cells in the thymus, cooperative antigen transfer is a very complex process that can be regulated at multiple levels. As suggested by the reviewer, the antigen transfer was shown to occur mostly by the scavenging of apoptotic bodies by thymic DCs (Perry et al, Immunity 2018). In this case, the increase apoptosis or autophagy of mTECs could indeed

effect TdTom transfer. To answer this question, we generated GSEA Enrichment plots of the genes associated with apoptosis and autophagy. As shown here, neither of the genes involved in the apoptosis or autophagy was changed after TLR9 stimulation of mTECs.

However, as previously shown, Aire⁺ mTECs further differentiate into “cornified” Involucrin⁺ post-Aire cells, which used the process of cornification as an alternative route to cell death (Michel et al., Journal of

Immunology 2017). Based on this, we included novel data showing that intrathymic TLR9 stimulation increased the number of Involucrin⁺ post-Aire mTECs in thymic medulla (see Supplementary Fig. 3e, f).

To answer whether MHCII molecules are also involved in a TLR-induced model of antigen transfer, we designed a new model system, in which eGFP-labeled MHCII (MHCII-eGFP knock-in mouse) is expressed only by a radioresistant thymic cell. As shown here, the surface eGFP is also efficiently transferred to Sirpα⁺ DCs and intrathymic injection of CpG ODN led to an increase in the amount of eGFP⁺Sirpα⁺ DCs. This suggests, that TLR/MyD88 signaling could also affect

the transfer of surface MHCII molecules to Sirpα⁺ DCs. Since most of these data shown in the manuscript has been focused on the specific aspects of mechanisms of cooperative antigen transfer, we think that this data is out of the scope of this manuscript. For this reason, we omitted the MHCII-eGFP model from the manuscript but show it here for review purposes only.

5. Aire⁺ mTECs are the major source of TSA and self-antigen transfers from these Aire⁺ mTECs to mDCs are crucial for negative selection of self-reactive thymocytes. It is not clear whether Aire-mediated TSA expression and transfer can also be modulated by TLR9/MyD88 signaling in mTECs?

- We thank the reviewer for this interesting question. We have included novel analysis comparing the representation of Aire-dependent and Aire-independent TRAs that have been found among the genes affected by TLR/MyD88 signaling. Now shown in Supplementary Fig. 2a, TLR/MyD88-induced/repressed transcripts were not enriched for Aire-dependent and Aire-independent TRA genes. This implies that TLR9/MyD88 signaling does not affect TRA expression in mTECs per se.

Unfortunately, due to the fact that there is currently no available model that would be suitable for studying the specific transfer of endogenous and physiological TRAs to thymic DCs, we cannot determine if TLR9/MyD88 signaling specifically affects the transfer of TRA. We can only speculate that since TLR/MyD88 signaling regulates the differentiation of mTECs^{high} into post-Aire mTECs, which serve as a reservoir of TRAs, and upregulate the chemokines that affect the migration of CD14⁺moDC into the thymus, is very likely that transfer of TRAs will be increased in response to TLR/MyD88 signaling.

REVIEWERS' COMMENTS:

Reviewer #1 (Remarks to the Author):

The authors greatly improved the manuscript and responded to all comments and concerns raised by the reviewers.

Reviewer #2 (Remarks to the Author):

The authors have addressed the vast majority of critical points that were raised by the referees, enabling them to significantly improve the manuscript

Reviewer #3 (Remarks to the Author):

The authors have responded to my comments satisfactorily.

One more suggestion: I'd like to see more speculations on the thymic endogenous ligands for TLRs on mTECs and their potential physiological significance. Would the self DNA fragments from apoptotic cells within the thymus serve as some of the ligands?

Point-by-point response to Reviewer's comments:

Reviewer #1 (Remarks to the Author):

The authors greatly improved the manuscript and responded to all comments and concerns raised by the reviewers.

Reviewer #2 (Remarks to the Author):

The authors have addressed the vast majority of critical points that were raised by the referees, enabling them to significantly improve the manuscript

Reviewer #3 (Remarks to the Author):

The authors have responded to my comments satisfactorily.

One more suggestion: I'd like to see more speculations on the thymic endogenous ligands for TLRs on mTECs and their potential physiological significance. Would the self DNA fragments from apoptotic cells within the thymus serve as some of the ligands?

- The question concerning the potential ligand(s) for TLR9 expressed by mTECs is indeed a very interesting topic for academic discussion. However, based on the data presented in our study, we can only suggest that it seems to be of thymic endogenous origin. We believe that any speculation regarding the nature of this ligand, at this moment, would be counterproductive as it can mislead and/or misinterpret data provided in this study. Such speculations would be more appropriate for a review article on a related topic. We are convinced that rather than speculation, rigorous, well designed, and focused experiments would help to elucidate the nature of such ligands.